

# Implementing northern peatlands in a global land surface model: description and evaluation in the ORCHIDEE high latitude version model (ORC-HL-PEAT)

Chloé Largeron[1,2], Gerhard Krinner[1], Philippe Ciais[2], and Claire Brutel-Vuilmet[1]

[1]CNRS and Univ. Grenoble Alpes, Institut de Géosciences de l'Environnement (IGE), 38000 Grenoble, France
[2]Laboratoire des Sciences du Climat et de l'Environnement, Institut Pierre-Simon Laplace, CEA-CNRS-UVSQ, CE Orme des Merisiers, 91191 Gif sur Yvette Cedex, France

*Correspondence to:* chloe.largeron@gmail.com

**Abstract.** Widely present in boreal regions, peatlands contain large carbon stocks because of their hydrologic properties and high water content, making decomposition smaller than primary productivity. We have enhanced the global land surface model ORCHIDEE by introducing northern peatlands. These are considered as a new Plant Functional Types (PFT) in the model, with specific hydrological properties for peat soil. In this paper, we focus on the representation of the hydrology of northern peatlands

and on the evaluation of the hydrological impact of this implementation. A prescribed map based on the inventory of Yu et al. (2010) defines peatlands as a fraction of grid cell represented as a PFT comparable to C3 grasses with adaptations to reproduce shallow roots and higher photosynthesis stress. The treatment of peatland hydrology differs from that of other vegetation types by the fact that runoff from other soil types is partially directed towards the peatlands (instead of directly to the river network). The evaluation of this implementation was carried out according to different spatial and temporal scales, from site evaluation to

larger scales such as watershed scale and all northern latitudes scale. The simulated net ecosystem exchanges are in agreement with observations from 3 FLUXNET sites. Water table positions were generally close to observations, with some exceptions in winter. Compared to other soils, the simulated peat soil have a reduced seasonal variability of water storage. The seasonal cycle of inundated peatlands area is also compared to flooded area estimated from satellite observations. The model is able to represents more than 89.5 % of the flooded areas located in peatlands areas, where the modelled extent of inundated peatlands

reaches 0.83 Mkm$^2$. However, the extent of peatland in northern latitudes is small enough that is does not impact the terrestrial water storage at scale of latitudes over 45 °N. Therefore, the inclusion of peatlands has a weak impact on the river discharge located in boreal regions.

## 1   Introduction

Peatlands are widely present in the northern latitudes and in permafrost regions. They contain large carbon stocks that are

estimated between 473-621 GtC in boreal regions (Yu et al., 2010). Due to high soil moisture and low winter temperature, the soil carbon is slowly decomposed and the accumulation of peat represents up to 25% of the total carbon pool in soils (Jobbágy and Jackson, 2000). Meanwhile, ongoing and projected climate change is particularly severe at these latitudes and leads to




thawing permafrost with increasing the active layer depth (Manabe and Stouffer, 1980; Cai, 2005; Collins et al., 2013). This large carbon reservoir may be partially mobilized. Peatlands are also one of the greatest natural sources of methane (Fung et al., 1991). Moreover, peatlands play an important role in the carbon cycle. Furthermore, the methane emission of peatlands depends significantly on the climate and especially on hydrological conditions at the surface. Considered as wetlands, these

areas have a significant influence on the surrounding climate through moisture and heat flux exchanges (Krinner, 2003; Pitman, 1991).

Given the importance of peatlands in the carbon and hydrological cycle, studies have attempted to include their representation in global models. The characteristics of peatlands concern as well vegetation, hydrology and carbon. As ecosystems, peatlands are very different than other land areas because the vegetation can survive in a permanently inundated area (Cronk

and Fennessy, 2001; Pezeshki, 2001). These conditions promote slower decomposition of carbon and the accumulation of carbon in the soils (Clymo et al., 1998). Peatlands are mainly located in high northern latitudes. Wide-spread initiation of peatland formation occurred when summer insolation started to decrease after about 11000 year BP (Tarnocai, 2006; MacDonald et al., 2006). High moisture content, a necessary condition for peatland development, is favoured by an important water supply and by the presence of frozen soil, which leads to reduced soil water infiltration. The large water holding capacity due to the hydraulic

properties of peat soil and the impermeable soils in a shallow depth due to frozen soils enhance a large peatland fractions in permafrost areas. However, the representation of peatland hydrology in large-scale models is complicated because peatlands remain sub-grid scale features conditioned by a range of complex processes.

The representation of peatlands in a land surface model requires to get a convenient biological, hydrological and carbon scheme adapted for peatlands. For this study, we use the global land surface model ORCHIDEE offline, which is part of the

IPSL coupled climate model.

In order to improve the $CO_2$ and energy fluxes of peatlands, the first step is to include a biological parametrisation of typical peatland vegetation as a new plant functional type (PFT), with peatland extent being prescribed.

Some studies have used the flooded area to model potential peat areas and then estimate their amount of methane emissions (Prigent et al., 2007; Ringeval et al., 2012). Although peatlands are characterized by high soil moisture content, the corre-

sponding water table depth (WTD) has a seasonal variation i.e. peatlands are not always flooded. Furthermore, most peatlands are located in high northern latitudes and undergo freeze and thaw periods, which have to be considered because they impact on the partitioning between runoff and infiltration.

In this study, we focus on a better representation of the hydrological processes occurring in peatlands located in northern latitudes. These developments were carried out in the high-latitude version of the model ORCHIDEE (Koven et al., 2009;

Gouttevin, 2012; Ringeval et al., 2012). A fixed map defines the location of peatlands, where each of them corresponds to a fraction of the grid cell. Peatlands strongly depend on local conditions and necessarily remain sub-grid features at spatial resolutions of the order of a hundred kilometres. This study therefore focuses on the global hydrological behaviour of the areas with high peatlands density. We evaluate modelled hydrological processes of peatlands against site and satellite observations and carry out sensitivity test of the water table depth as a function of the spatial density of peatlands.



## 2 Model description

### 2.1 ORCHIDEE High Latitude version

In the present study, we introduce the northern peatland (version ORC-HL-PEAT, rev. 3058) in a specific high-latitude version
(ORC-HL/MICT v1) (rev. 1255) (Koven et al., 2009) of the IPSL land surface model ORCHIDEE (Krinner et al., 2005) that

includes a soil-freezing scheme (Gouttevin, 2012), which is crucial to represent high latitudes land surface processes.

We run the model offline, driven by a prescribed atmospheric forcing rather than coupled with the atmospheric model, in
order to facilitate the assessment of the newly introduced processes, which will be described in the following. In addition to the
meteorological driving data, the model requires spatialised parameters such as vegetation distribution, soil texture, topography
and watershed location to represent land surface properties.

Including bare soil, there are 13 different plant functional types (PFTs) in the model. In this study, dynamic vegetation is not
activated and the fraction of each PFT is prescribed. We use the Olson et al. (1983) vegetation map, which defines 94 vegetation
classes. These are then converted to 13 PFT fractions at a resolution of $0.5°$.

Some hydrological variables such as transpiration and interception vary as a function of the vegetation. The transport of
water in the soil is described by the 11-layer scheme of De Rosnay (1999), which calculates heat and moisture transport

between each soil layer. The water balance of the soil is defined separately as a function of class of vegetation and clustered
for bare soil, trees and grasses. Each of these three soil types has a separate water balance. The fraction of the area of each
soil type is calculated as a function of the fraction of the area of the corresponding PFT. However, the soil porosity is defined
only as a function of the dominant soil texture in the grid cell, based on textural classification data from the global Food
and Agriculture Organization map (FAO, 1978). Only one soil parameter is defined per grid cell, which describes hydraulic

conductivity, residual and saturated water content as well as the Van Genuchten parameters which describe the hydrological
properties of the soil.

The runoff and drainage transport to the river and oceans are accounted for by a routing scheme which are separated into 3
reservoirs with different velocities (Fekete et al., 1999; Vörösmarty et al., 2000; Ngo-Duc et al., 2005). To simulate peatlands
processes, the routing scheme is activated. The model runs at a half-hourly time step. The processes are simulated using a daily,

monthly or annual time step, depending on the process involved.

### 2.2 Model experiments

The meteorological forcing data used to drive ORCHIDEE model were taken from CRUNCEP v5.3 (Viovy and Ciais, 2011).
This forcing is a combination of the 6-hourly climate forcing data NCEP corrected by the monthly observations data set of
the Climatic Research Unit over the period 1901-2013 (Saha et al., 2014; Harris et al., 2014). The meteorological forcing data

WFDEI is also used to determine the sensitivity of the input data as the amount of precipitation (Weedon et al., 2014). This
forcing is based on the climate reanalysis ERA-Interim on a 3-hourly time step corrected with the observed precipitation data
of Global Precipitation Climatology Center with the time series of 1979-2013. Both simulations were carried out over north of
$45°$ North at $0.5°$ resolution.



The simulations were performed at a 30 min time step with a spin up of several hundred years to ensure that hydrological processes have reached equilibrium.

We evaluate the modelled processes at different spatial scales in order to evaluate the hydrological behaviour on peatland sites, and also to evaluate the impact of the inclusion of peatlands on large-scale hydrology, which must be carried out for

global climate models.

Firstly, we evaluate the modelled peatland processes at site measurements with using the FLUXNET meteorological database (Baldocchi et al., 2001) for the peatland sites of Degero, Fajemyr and Siikaneva (Lund et al., 2009; Rinne et al., 2007; Nilsson et al., 2008; Sagerfors et al., 2008). The advantage of using the FLUXNET database is that it provides continuous meteorological data such as precipitation and temperature, which help to compare data of net ecosystem exchange (NEE) or water table

depth (WTD).

The site of Fajemyr is located in the southern Sweden (56°15' N, 13°33' E). It is an ombrotrophic bog in a temperate climate at 140 m altitude. The annual mean (1961-1990) temperature and precipitation are 6.2°C and 700 mm respectively. The WTD is generally below the surface where many mounds can be seen on the surface with moss, sphagnum, sedge vegetation and small shrubs (Lund et al., 2009).

The site of Siikaneva is located in Ruovesi in the south of Finland (61°49' N, 24°11' E), in the southern boreal zone where the temperature varies on average from -8°C to 16°C. The annual mean (1971-2000) temperature is 3.3°C and the annual mean precipitation 713 mm. The year 2005 used in this study has a higher amount of precipitation than average. During spring the snow and peat melt in the beginning of April and the permanent snow cover starts early December. Dominated by shallow vegetation, this minerotrophic fen is populated by sphagnum and peat mosses. The surface is relatively flat with no pronounced

hollows and bumps (Rinne et al., 2007).

The site of Degero is located in the county of Vasterbotten in northern Sweden in the middle boreal zone (64°11' N, 19°33' E). The climate in this site is temperate, cold and humid. This site of peatlands is a boreal oligotrophic minerotrophic fen surrounded by forest with herbaceous vegetation and an average depth of peat between 3-4 m. Located at an altitude of 270 m, it covers an area of 6.5 km$^2$ (Schubert et al., 2010). The annual mean temperature is 1.2°C and annual mean precipitation is

567 mm (1971-2000) (Laine et al., 2011).

The data obtained for these 3 sites are based on available FLUXNET data from 2004 to 2005 for Siikaneva, from 2001 and 2005 for the Degero site, and from 2005 to 2006 for the Fajemyr site (Baldocchi et al., 2001).

The site evaluation of the hydrology of peatlands was carried out using the WFDEI meteorological forcing with 0.5° of resolution, which contain the geographical location of the site data. The WFDEI forcing was chosen since the precipitation

data were corrected from the observations. The evaluation of the impact of the inclusion of peatlands on a larger scale were performed with both CRUNCEP and WFDEI forcing.

The years where the measurements of WTD coincide with the available years of FLUXNET meteorological forcing at these sites is chosen and correspond respectively to the year 2002, 2005 and 2006 for the Degero, Siikaneva and Fajemyr sites. For this evaluation, the corresponding grid cell is assumed to be covered by 100% of the PFT of peatlands.





### 2.3 Modelling peatlands

The inclusion of peatlands processes requires locating them. Here, we choose to use a fixed map of peatland rather than a model that describes only inundated areas as it already has been studied in ORCHIDEE (Ringeval et al., 2012).

The structure of the hydrological scheme in ORCHIDEE implies that the peatlands have to be considered as linked to a new

Plant Functional Type (PFT) with adapted biological parameters; only this allows a separated calculation of the water balance.

#### 2.3.1 Peatlands map

Given the difficulty of locating large-scale peatlands, many methods exist to define these surfaces globally, but all of them have some biases. The TOPMODEL approach, which determines areas where the soil is temporarily saturated, does not allow restoring the hydrology of peatlands; these are then described as wetlands (Ringeval et al., 2012). Other studies such as Wania

et al. (2009a) use the IGBP-DIS soil map to identify peatlands with regions with high soil organic carbon content, including other landforms with high soil organic carbon content such as yedoma (Tarnocai et al., 2009; Zimov et al., 2006).

To represent the evolution of the hydrology of peatlands over time, we chose to locate peatlands without integrating other wetlands or other soils with high carbon content. We therefore use the peatland map obtained by Yu et al. (2010). This global map at a resolution of 2 km recognized peatlands where a grid cell contains at least 5% of peatlands. These data are ob-

tained from inventories by countries, or from gleysols and histosols from the Harmonized World Soil Database V1.1 when the inventory is missing (Yu et al., 2010).

We constructed a new land cover map including peatlands as a combination of the original 13 PFT map and the global peatland map from Yu et al. (2010), with peatlands considered as a $14^{th}$ PFT. The peatland map of Yu et al. (2010) is interpolated onto a $0.5° \times 0.5°$ regular grid in order to obtain a fraction of peatland $f_{peat}^{Yu}$ for each grid cell. The fraction of the new PFT

$f_{\text{PFT } 14}$ of peatlands is then corresponding to $f_{peat}^{Yu}$ limited to the maximum fraction of grassland. By conservation, the fraction of grassland is reduced by the amount of the peat fraction.

This method leads to a reduction of 19% of grassland area north of 45°N. The resulting peatland area located in latitude between 45° and 90° north is 3.8 Mkm$^2$, which is in adequacy with the 4.0 Mkm$^2$ estimated from observations (Tarnocai et al., 2009). This corresponds to 52.1% of the original global peatlands map where 76.8% is located north of 45°N.

#### 2.3.2 Biological processes

Mosses, sphagnum and grassland mainly compose the vegetation in peatlands. These vegetation types can be grouped as flood tolerant C3 graminoid and sphagnum (Wania et al., 2009b). Assuming peatlands have similar biological properties as C3 grasses determined by Krinner et al. (2005), the $14^{th}$ PFT added to represent peatlands has the same prescribed basic biological properties as the C3 grasses PFT.

In peatlands, the vegetation can survive in saturated areas. In the model, the parameterisation of water stress is such that stress only exists in case of lack of water. Thus, since they do not suffer from inundation stress, all PFTs represented in the baseline version of ORCHIDEE are flood tolerant. This parameterisation remains unchanged for peatlands.

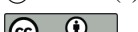



On an energetic level, peatlands have a primary production lower than the grasses. The vegetation in peatlands has a nutrient deficiency caused in part by high water content, resulting in lower plant growth (Bridgham et al., 1996). In peatlands, the nitrogen and phosphorous limitation leads to a reduced photosynthetic capacity (Cronk and Fennessy, 2001; Pezeshki, 2001). Observations showed lower leaf nitrogen concentrations in peatlands than in other terrestrial plants in the same group causing
a lower concentration of Rubisco (Aerts et al., 1999).

Because the ORCHIDEE high-latitude version (ORC-HL) model does not include the nitrogen cycle, the lower NPP observed in peatlands due to the nitrogen limitation has to be taken into account by strengthening RuBisCo limitation on carboxylation, which allows the fixation of carbon dioxide $CO_2$ in the biomass. The gross primary productivity (GPP) is influenced by nutrient availability, climate conditions such as sunshine, relative humidity, temperature and $CO_2$ concentrations in the
atmosphere and by the maximum rate of carboxylation $V_{cmax}$, which corresponds to the maximum rate of RuBisCO.

The lower productivity in peatlands is represented empirically by reducing $V_{cmax}$ by a factor $\delta_N$ corresponding to the nitrogen limitation. This limiting factor $\delta_N$ is chosen in such a way that the amplitude of simulated daily mean GPP is consistent with the observed daily mean GPP. For this study, the observation site chosen to adjust the GPP is the Swedish peatland site of Degero, which is a fen covered by herbaceous vegetation (Schubert et al., 2010). After applying an appropriate limiting factor,
the simulations are forced by meteorological data of Degero using FLUXNET database (Baldocchi et al., 2001). The limiting factor $\delta_N$ induced a value of $V_{cmax}$ applied for peatlands of 16 $\mu$mol $m^{-2}s^{-1}$, instead of the value fixed in ORCHIDEE of 70 $\mu$mol $m^{-2}s^{-1}$ for non-peatland herbaceous C3 plants. This parameter setting has been tested with other peatlands sites (Fajemyr, Minnesota, Siikaneva which are located in Sweden, Minnesota State of America and Finland respectively (Lund et al., 2009; Shurpali et al., 1995; Rinne et al., 2007)), for which the estimated GPP is also improved.
The reduction of the maximum rate of carboxylation leads to a lower Leaf Area Index (LAI), which also affects the water balance. In the model, the roots are represented by a root vertical profile $R(z) = \exp^{(-\alpha z)}$. The vertical distribution of root biomass is described by a decay parameter $\alpha$ (m$^{-1}$), which is determined for each vegetation type in order to obtain adequate rooting depths and allows a good estimation of water flows.

Frolking et al. (2001) studies estimated the rooting depth of peatlands (bogs as well as fens) around 30 cm. The corresponding
value of the decay parameter adapted for peatlands is $\alpha$ = 10 m$^{-1}$ where 99 % of the density of roots is above 30 cm of depth, instead of the usual ORCHIDEE value of 4 m$^{-1}$ for herbaceous vegetation. The reduction of the rooting depth could increase water stress in dry periods, when the surface layers of soil has a water deficit.

### 2.3.3   Hydrological processes

Peatlands have to be represented with a specific hydrological scheme. The low water flows of peatlands lead to high water
content in these soils, which are partly maintained by poor drainage. In the model, the water balance is described for groups of plants. In the original version of ORCHIDEE, three different water columns are defined. These columns pertain to bare soil, trees and grasses, respectively. To represent the hydrological conditions of peatlands, we have added a new water column corresponding to the peatland PFT. Specific hydraulic properties are applied to this soil column to improve the representation of peatlands.





Peatlands are characterized by a higher average soil moisture content. Peatland lateral water inflow comes from precipitation, surface runoff and from nearby water tables. To represent these processes, we choose to infiltrate into the soil column of peatlands the runoff coming from the soil columns regrouping all the non-peatlands PFTs in the same grid box.

This water supply is gradually infiltrated into the deeper soil layers. This quantity of water can be evacuated by the outflow where the infiltration speed depends on the hydraulic conductivity of each layer as described by De Rosnay (1999). In peatlands, the water content is mainly maintained in soils due to poor drainage and is considered as negligible (Boatman and Tomlinson, 1973). To prevent water lost by the drainage, we choose to block the deep drainage at the deepest soil layer because peatlands usually have no deep drainage.

In the peatland scheme, the amount of standing water above the soil surface is taken into account as an additional reservoir. When the water content of the first layer exceeds the saturated water content $\theta_s$, the reservoir is filled by the additional quantity of water which cannot be infiltrated in the soil. This amount of water is then infiltrated into the soil as soon as the soil becomes under saturated. The maximum content of this reservoir is 10 cm, in accordance with observations (Booth et al., 2005). When the capacity of the reservoir is exceeded, the surface runoff occurs.

The water supply of peatlands is produced by the runoff from other soils in addition to precipitation. By conserving the mass of water, the amount of water needed to maintain the soil close to the saturation increases with the fraction of peatlands. This choice leads to a dependence of the water supply as a function of the fraction of peatlands in a grid cell.

A study case with the Sweden site of peatlands at Degero (Schubert et al., 2010) where the meteorological forcing of the year 2001 showed that the greater the fraction of peatlands in the grid cell, the smaller the amount of water coming from the runoff from other soils. This dependence implies higher total soil moisture for low fractions of peatlands, caused by the routing of runoff from other soils to peatlands. Conversely, peatlands fractions above 80 % have a slightly higher amount of water than the intermediate fractions due to the storage of peatland runoff in the standing water reservoir, which is then re-infiltrated into the peat soils (shown Fig. 1). The distribution of the fraction of the non-peat PFTs in a grid cell does not substantially influence this result (tests performed but not shown).

The hydrological variations of peatlands are assessed using the water table depth (WTD). Perched water tables within peatlands are unusual (Shi et al., 2015). Therefore the multi-layer hydrological scheme (De Rosnay, 1999), developed for standard soils, is not necessarily well adapted for the simulation of peatland hydrology. We thus diagnose the simulated peatland water table as in a bucket system as a function of the total amount of soil water in the peat column.

The amount of water evaporated depends on the water availability of the soil. The evaporation $E$ is limited by a set of factors $\beta$, such as the evaporation capacity of the soil $\beta_{evap}$, its transpiration capacity $\beta_{transpir}$, and its interception capacity $\beta_{inter}$, as shown in Eq (1) (Budyko, 1961; Farquhar et al., 1980; Ball et al., 1987).

$$E = \beta E_{pot} \tag{1}$$
$$\beta = \beta_{evap} + \beta_{transpir} + \beta_{inter} \tag{2}$$





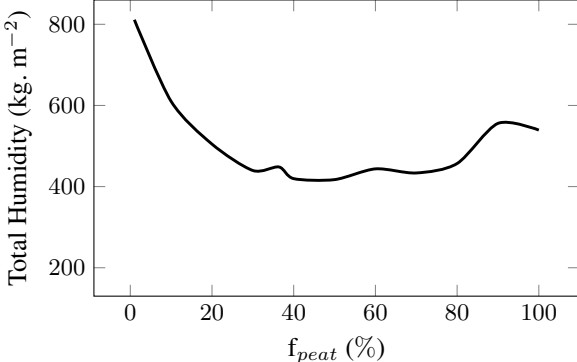

**Figure 1.** Amount of total humidity contained in the column of peat soil at the Degero site in 2001 as a function of the percentage of peatland imposed in a grid cell.

Peatland soils are flooded most of the year with a vegetation such as mosses that are saturated with water. In that case, the relative humidity of the air at the surface of the soil is close to saturation. The calculation of evaporation is defined with the air humidity at 2 m high above the surface per grid cell, taking into account the water balance of peatland as well as other PFTs. This unique, grid-cell average air humidity is typically lower than the air humidity at the surface of flooded areas. To address

this problem, we add a resistance to evaporation $\beta_{evap}$ to further limit the evaporation at the soil surface of peatlands. This parameter is applied for all PFTs. In order not to affect other soils, this resistance is applied to the calculation of evaporation once separated by soil column.

We choose a reduction R in accordance with the soil humidity. Since this amount of water increases with the fraction of peatlands in a grid cell $f_{peat}$, the evaporation resistance factor $R$ is applied as a function of the fraction of peatlands following

Eq. (3). This reduction avoids the overestimation of evaporation and has been calibrated so that the modelled latent heat flux corresponds to the observation data at Siikaneva, using the meteorological forcing of this Finland peatlands site (Rinne et al., 2007).

$$R = \frac{f_{peat}}{2} + \frac{1}{2} \tag{3}$$

The flow of transpiration by the vegetation of peatlands is also reduced by reducing the gross primary productivity with the

parameter $V_{cmax}$. In peatlands, this lower rate can also be explained by the underestimation of the modelled air humidity at the canopy.

## 3 Results

### 3.1 Site evaluation

The mean diurnal cycle of net ecosystem exchange (NEE) of the modelled peatlands PFT is compared to observations with a

10-day running mean smoothing that eliminates day-to-day variations. This allows evaluating the simulated evolution of the





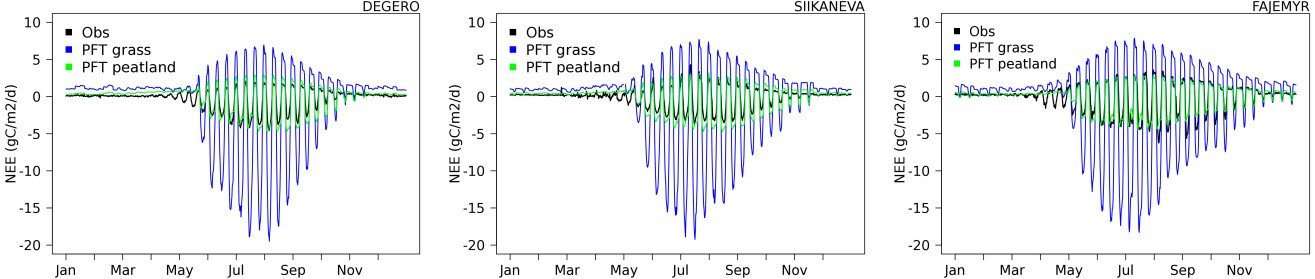

**Figure 2.** Diurnal cycle of NEE smoothed with a 10-day running-mean filter of modelled PFT peatland (green), modelled PFT grass (blue) and observation (black) from the peatlands sites of Degero (left), Siikaneva (center) and Fajemyr (right).

diurnal cycle of NEE on seasonal time scales (Fig. 2). This profile is based on the average of years available for each peatland sites. The reduction of the prescribed maximum carboxylation rate, carried out in order to correct the overestimated gross primary productivity in the case of peatlands vegetation, leads to a lower amplitude of the diurnal cycle of the NEE.

The simulated amplitude of the maximum net uptake of $CO_2$ by grasses (i.e. non-peatland vegetation simulated on normal
ground) reached 19 gC/m$^2$/d for all of the 3 different sites during the summer, while this amplitude cannot reach more than 4.8 (4.7; 4.5) gC/m$^2$/d at the Degero (Siikaneva; Fajemyr) site both in observations and in the peatland simulations. That is, the capacity of daily carbon exchange is 4 times lower for peatlands than for grasses under the same climatic forcing. The model underestimates the maximum net peatland uptake by between 10 and 35% for the Degero and Fajemyr sites and overestimated by about 10% for the site of Siikaneva. The daily maximum deviation is on average of 15, 16 and 20% during June and
September. The profile of modelled peatland is relatively well represented and the correction of the productivity leads to a good representation of the seasonal and diurnal variability of observed NEE of peatlands.

The hydrology of peatlands is evaluated by comparing the modelled and observed WTD as shown on Fig. 3. The modelled water table is driven by the meteorological data given at these sites, where the monthly precipitation and temperature are shown at the top of the Fig. 3. The precipitation counts both rainfalls and snowfalls, converted to mm.
In the setup of site simulations, the amount of water in the modelled peat soil is filled only with precipitation. However, the water supply of the minerotrophic sites such as Degero and Siikaneva also comes from drainage and sub-surface runoff of other land scape elements. This phenomenon cannot be represented in the model since the amount of water coming from sub-surface runoff and drainage remains unknown. The modelled WTD therefore rather directly follows the quantity of precipitation. Model results for the minerotrophic sites (Degero and Siikaneva) show that the water supply from precipitation is enough
to reproduce the observed water table position. For these 3 sites, the modelled WTD is in agreement with the observations during the summer, when the soil is no longer frozen. Results from the Degero fens site underestimates the WTD during the summer. This bias may be explained by the amount of water from groundwater which is not represented in the model. The



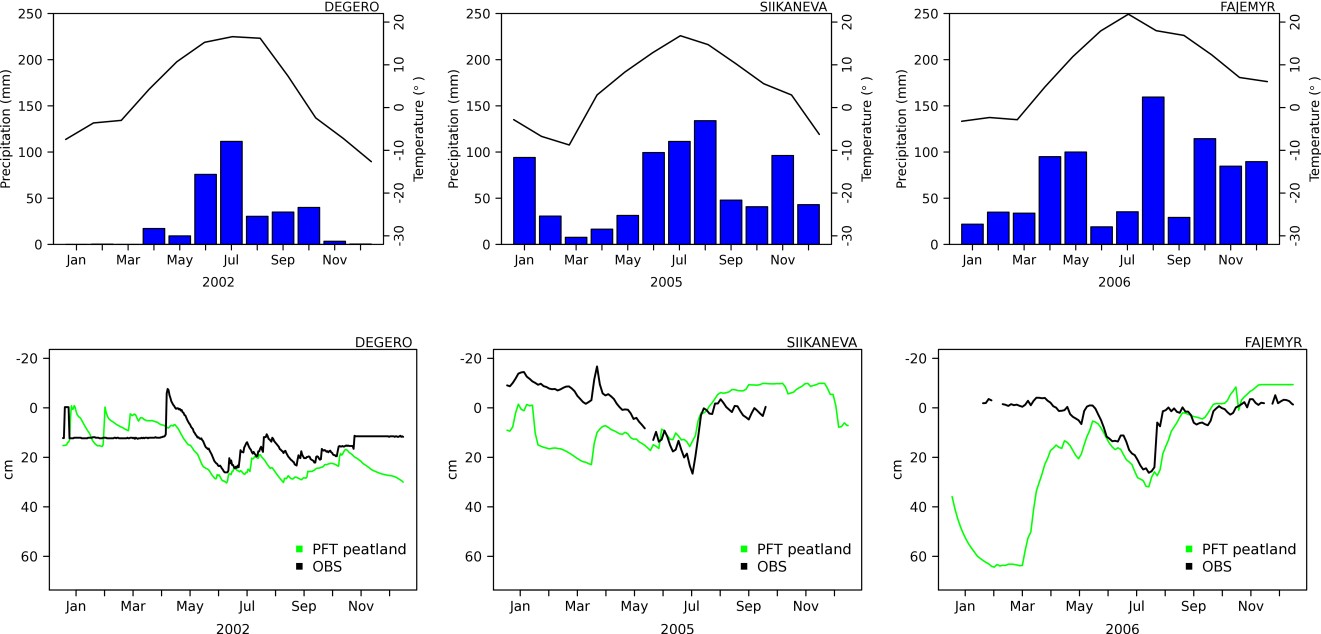

**Figure 3.** Monthly mean temperature (black line) and monthly precipitation (blue bars) at the top and seasonal cycle of modelled (green) and observed (black) water table depth at the bottom for the peatland sites of Degero (left), Siikaneva (centre) and Fajemyr (right).

opposite is observed at the Siikaneva site and may come from an outgoing flow as a small drainage rate. The modelled WTD is underestimated when the soil is frozen. Since the infiltration of snowfall cannot occur, the water content is underestimated.

The modelled WTD at the Fajemyr site reached 64 cm during March 2006. This value can be explained by a low rainfall in the previous year where the annual precipitation is under 75 % of the amount of the year 2006 with a monthly value less than 50 mm/month between September and December.

The meteorological conditions of the minerotrophic peatlands allows a better representation of the hydrology of peatlands than the ombrotrophic bogs such as Fajemyr, where during February and March the simulated frozen soil prevents infiltration and thus maintains undersaturated soil conditions.

In an experiment where we add water content to force saturated soil conditions below 30 cm depth, the model simulates a WTD for the Fajemyr sites that matches the observed water table even in winter.

## 3.2 Sensitivity of types of peatlands to precipitation

The misfit between modelled WTD and site measurements could be caused in part by the unknown amount of water from runoff which is not represented on site-simulation where peatlands represent 100 % of the grid cell. Since we cannot separate bogs and fens at the spatial scales relevant here, we consider in the model that all peatlands are fed by runoff. Here, bogs are





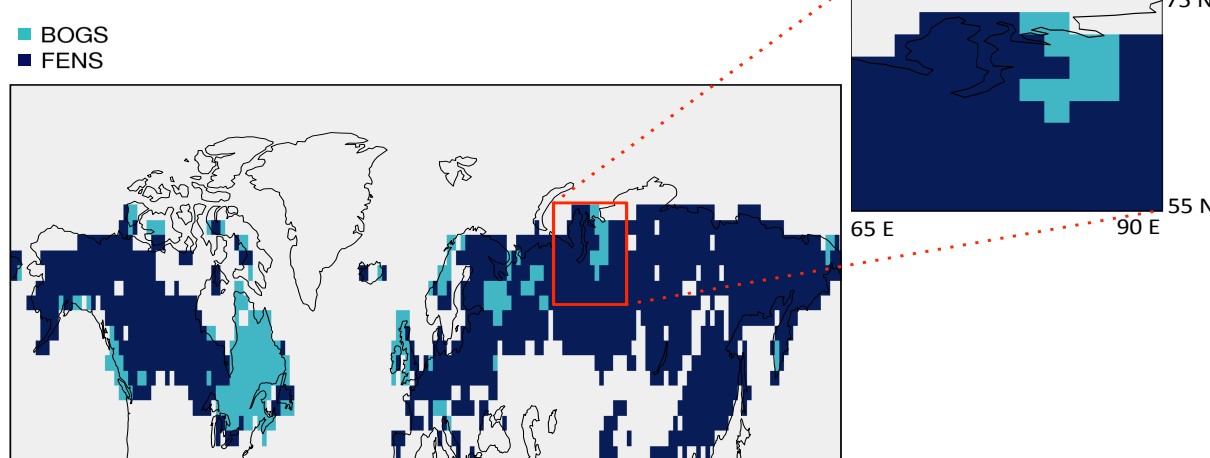

**Figure 4.** Map of northern peatlands separating the type of bogs (light blue) from fens (dark blue) modelled peatlands based on conditions of the water table depth fed by the precipitation only.

distinguished from fens from the WTD when peatlands are fed only by precipitation. A precipitation sensitivity study of the different types of peatlands is carried out by modifying the precipitation according to different multiplicative factors.

We assume that ombrotrophic bog areas, i.e. peatland fed only by rainfall that do not receive input from other soil columns, correspond to regions of peatlands that have a mean WTD less than 30 cm (in accordance with observations (Booth et al., 2005)

) at least 4 consecutive months in the mean year, for the 1990-2010 period, with a water supply coming only from precipitation. Usually this condition tends to be fulfilled between January and April. By deduction, other peatlands present in the Yu et al. (2010) map are considered to be minerotrophic fens because they require additional water input to maintain a shallow WTD throughout the year.

The ombrotrophic bogs are diagnosed as localised in areas where peatlands are flooded during the summer. These bogs are

located in north-eastern Canada, on the west coast of Canada, central Russia, United Kingdom, Norway and north-west Russia near the White Sea (represented in light blue in Fig. 4 ). The total area of these simulated ombrotrophic peat bogs represents more than $0.5$ Mkm$^2$ among the $3.8$ Mkm$^2$ of the northern peatlands over 45°N. However, the large-scale climate forcing does not allow to represent sub-grid conditions (micro-climates, topographic setting, etc.) that occur especially in mountain regions and as local sources of water from upslope areas. As a result, the regions which we identify, on large scales, as favorable for

the occurrence of ombrotrophic bogs (regions in light blue in Fig. 4) can of course also contain minerotrophic fens.

On a regional scale, we carried out a sensitivity study of the simulated peatland hydrology to precipitation. The selected study area concerns western Siberia between 64-90°E and 55-75°N (selected zoomed regions in Fig. 4). In parts of this region, large-scale conditions are favorable for the occurrence of ombrotrophic bogs, but in other parts of this region, peatland occurrence requires complementary horizontal water input in addition to the large-scale climate forcing. In this region, the



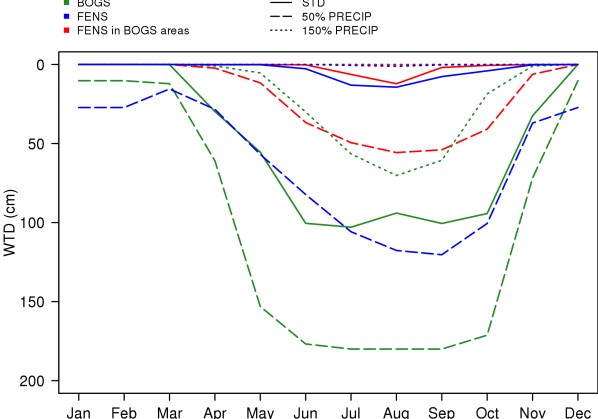

**Figure 5.** Seasonal cycle of WTD of minerotrophic (FENS), ombrotrophic (BOGS) and minerotrophic in regions localised as ombrotrophic (FENS in BOGS areas) with the standard precipitation (STD) (full line), 50% of the precipitation (50% PRECIP) (dashed line) and 150% of the precipitation (150% PRECIP) (dotted line)

annual precipitation varies between 400 and 500 mm/year from 1990 to 2010. The forcing of the same year has been repeated for 5 consecutive years by varying the prescribed precipitation rate between 50% and 150% around the true value. We choose the year 2000 which is a year with an average precipitation of 450 mm/y, where the inter-annual variability of the precipitation is 12% of the mean precipitation in this region.

Snowfall in these Siberian areas covers the soil with water from December to April, while the averaged simulated water table depth in summer reaches more than 1.5 m depth for ombrotrophic bogs soils that are not supplied with water by runoff (as shown in Fig. 5). In the case of ombrotrophic bogs, the 50% precipitation reduction results in a summer water table depth increase of 93% . When precipitation is increased by 50 % the WTD in summer is reduced by 37% (Fig. 5).

The mean WTD of soil of minerotrophic fens reaches 14 cm in summer. The water supply from the runoff allows the fen
soils to have a WTD closer to the surface than bog soils. When we reduce the precipitation by 50%, the water table depth of fen soils increases by 1 m, which corresponds to a relative increase of 7.19 (Fig. 6). The additional water supply makes minerotrophic peatlands (FENS) much more sensitive to precipitation than ombrotrophic peatlands (BOGS) (as shown in Fig. 6). A 50% precipitation increase leads to a permanent flooding of these soils.

However, minerotrophic peatlands in this study are located south of 64°N. In order to compare sensitivity to precipitation
of minerotrophic with ombrotrophic peatlands in the same weather conditions, it is advisable to consider also minerotrophic peatlands in areas where peatlands have been diagnosed as ombrotrophic. To make this comparison, a third simulation was carried out by feeding the peatlands in the ombrotrophic regions by runoff ("FENS in BOG areas" case in Figure 6). The WTD of these soils are equivalent to other minerotrophic soils. The reduction of precipitation by 50% results in a smaller increase of WTD than in the region of peatlands defined as minerotrophic, which reaches 55 cm depth in the summer instead of 12 cm.



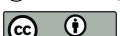

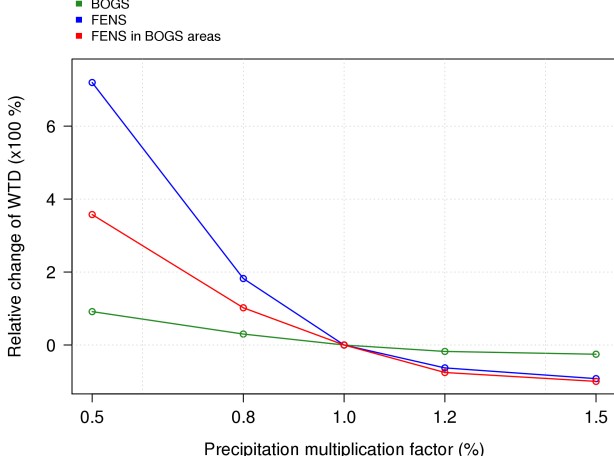

**Figure 6.** Relative change of water table position of peatlands in August as a function of precipitation multiplication factor ranging from 0.5 to 1.5 depending on the type of peatlands: minerotrophic (FENS), ombrotrophic (BOGS) and minerotrophic in regions localised as ombrotrophic (FENS in BOGS areas).

To summarize, our simulations show that minerotrophic peatlands are more sensitive to precipitation than ombrotrophic peatlands. The reduction of the precipitation by a factor of two leads to a rise of the WTD up to 8 times deeper for minerotrophic soils while this change does not exceed 2 times deeper for the ombrotrophic soils. This sensitivity is also seen when the runoff process is applied to areas defined as ombrotrophic (case "FENS in BOG areas").

## 3.3 Large scale hydrological impact

The inclusion of peatlands in the ORCHIDEE land surface model had to be assessed on a larger scale in order to determine the influence that peatlands have on large-scale hydrology in northern latitudes. After evaluating the processes of simulated peatlands on measurement sites, this study evaluates the impact of peatland implementation into the model on the flow of rivers in boreal watersheds and on the water mass changes of northern latitudes studied at different time scales.

### 3.3.1 Impact on the river discharge

The transport scheme of ORCHIDEE (Ngo-Duc et al., 2005) stores the water from the runoff and the drainage in 3 reservoirs with different residence times. Since the implementation of the peatlands in the model leads to the redirection of runoff from the other soil columns to the peat soils, one might expect some impact on the simulated river discharge. This impact is evaluated here for the Ob basin, which represents one of the largest boreal basins (above 45 °N). This watershed is located in abundant peatland areas, particularly north of 60°N. Although the average percentage of peatlands remains less than 10% per grid cell of 0.5° of resolution, more than half of the grid cells have a non-zero fraction of peatlands. Above 60°N, peatlands are present on more than 96% of the grid points of the Ob basin.

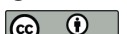



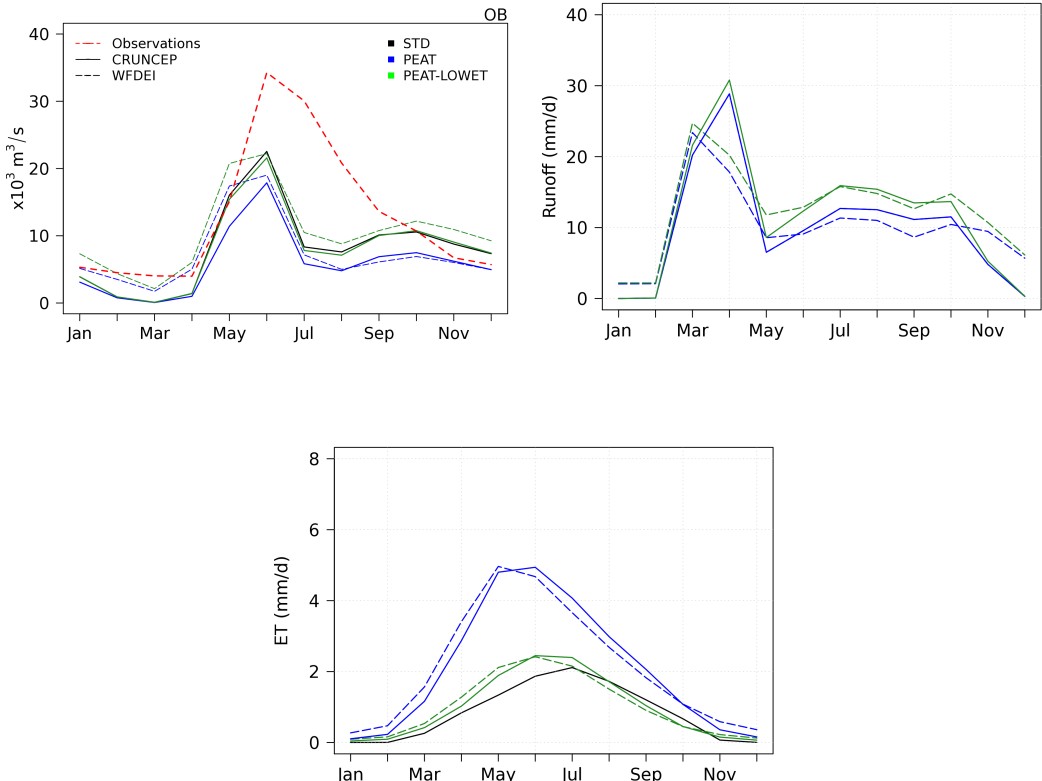

**Figure 7.** Mean annual cycle of river discharge (top), runoff of peat soils (center) and evapotranspiration of peatlands (bottom) for the Ob river basin. The observed river discharge is represented in red. The simulation are given with the standard version of ORCHIDEE (STD) (black), with peatlands scheme without (PEAT) (blue) and with the reduced (PEAT-LOWET) (green) evaporation.

We compare the modelled river discharge of the Ob basin in 3 different simulations. The standard simulation (STD) corresponds to the version of ORCHIDEE which includes soil freezing (Gouttevin, 2012) and excludes the peatland scheme. In addition, two peatland configurations are shown: PEAT and PEAT-LOWET which represents respectively the simulations where peatlands are activated with a standard and a modified evapotranspiration scheme where the evaporation is reduced by
5   R factor as in Eq. 3.

Mean annual cycles of monthly mean discharge are computed with the CRUNCEP and WFDEI meteorological forcing at 0.5° of resolution, for the 1979-1994 period for which we have observations. We carried out these simulations with different forcing files in order to study the sensitivity of the results to the meteorological forcing, but we will first show results obtained with the CRUNCEP forcing. These simulations are compared with observation from the Global Runoff Data Centre (shown in
10   Fig. 7) (Fekete et al., 1999).





The modelled river discharge with the original ORCHIDEE version underestimated the river flow of watershed located in boreal regions. This underestimation is known with the version of ORCHIDEE that does not include the snow scheme by Wang et al. (2013) and comes from the overestimate of snow sublimation (Ringeval et al., 2012; Wang et al., 2013). Here, the objective is to evaluate the impact of the inclusion of peatlands on the river flow compared to the same version of the model that does not include this scheme.

The modelled mean seasonal cycle of the river discharge for the boreal Ob watershed is shown in Fig. 7. The modelled river discharge PEAT with the peatland scheme results in a 20% reduction in the maximum river discharge obtained by the standard ORCHIDEE High latitude version (STD) (Fig. 7). The reintroduction of runoff into peat soils allows higher soil moisture resulting in an increased evapotranspiration, which can reach 4.9 mm/d for the average month of June instead of 2.1 mm/d on average for the standard version in the Ob watershed. The increase in evapotranspiration results in a difference in river flow of more than 4600 $m^3$ $s^{-1}$ on average over the mean month of June.

The simulated river discharge PEAT-LOWET, which includes the reduction applied as a function of the water supply (translated by the fraction of peatland in a grid cell), slightly reduces the underestimation of the river flow of the boreal basins. The average evapotranspiration of Ob basin peat soils reaches an average of 2.4 mm/d in the mean June, which approximately corresponds to the values obtained with the STD version. The reduction of evaporation leads to an increase of surface runoff by 577 mm/an on average of all peat soils in the Ob basin. The peatlands scheme has a negligible impact on the river discharge when the reduction of evaporation is applied.

These results are not very sensitive to the meteorological forcing used. The seasonal peak of runoff from peat soils occurs one month earlier with the WFDEI than with CRUNCEP forcing. This results in a large river flow that occurs earlier in the season with the WFDEI forcing. The behaviour of the modelled river discharge of the Ob basin is similar for both meteorological forcings after the month of June.

In all cases, the introduction of the peatland scheme does not alleviate the underestimate of the spring peak and summer discharge. This tends to confirm that this underestimate, at least the underestimate of the springtime maximum linked to snow melt, is due to the known overestimate of snow evaporation mentioned before.

### 3.3.2 Impact on terrestrial water storage

We now compare the simulated total terrestrial water storage (TWS) variations north of 45°N with (simulation PEAT) and without (simulation STD) the peatland scheme with the satellite observations from the Gravity Recovery and Climate Experiment (GRACE) mission (Tapley et al., 2004) at different time scales.

GRACE provides estimates of mass anomalies by measuring the variations of the geopotential field between the two satellites. These estimates quantify changes in the water mass of land surfaces, ice and oceans. The GRACE data are available from April 2002 until June 2014. They are interpolated in a grid cell of 1°x1° resolution with monthly data of total water mass variations ΔTWS, which is defined as the difference between the total water mass obtained at the time of the TWS measurements and the average total water mass between January 2004 and December 2009. In this study, we use GRACE release 5 spherical harmonic coefficients from the Jet Propulsion Laboratory (JPL-RL05) available from http://podaac.jpl.nasa.gov/grace.



**Figure 8.** (a) Land surface above 45 °N terrestrial water storage and the contribution of each reservoir in the model. Terrestrial water storage (TWS) variations of latitudes over 45 °N from GRACE (grey), the STD (black lines) and from the PEAT version (black dashed) for all peatlands grids cells with both peat and non-peat soil columns, including the water reservoir variations of total humidity of the soil (b) and the variations of the total humidity of the peat soil column only (c).

The observed TWS are compared with the simulated total water storage calculated from the water reservoirs represented in ORCHIDEE: surface runoff (FAST), deep drainage (SLOW), lateral flux (STREAM), floodplains (FLOOD), snow mass (SNOW) and humidity of the soil (SOIL).

The top panel of Fig. 8 shows the simulated area-average TWS variations by the STD version of the model between 2002
5   and 2013 and the contribution of each modelled water reservoir north of 40°N. The modelled TWS variations represent only 73% of the variation of TWS observed by satellites over all latitudes polewards of 40° North. The mean amplitude of the





simulated TWS variation is 6.8 cm instead of the 9.4 cm observed with GRACE. Maximum TWS occurs during spring and rapidly decreases when the snow melts.

At a seasonal scale, the negative contribution of the modelled variation of TWS occurs too early compared to the observations (Fig. not shown). This trend is due to the same shift of the contribution of the modelled TWS due to the snow where the melting of snow is anticipated. The inter-annual variability of the modelled TWS of all grid cells of the northern latitude is under-estimated compared to the satellites observations (Fig. 8).

The seasonal and inter-annual variation of TWS in boreal regions is mainly affected by the changes in snow mass and changes in water contained in soils. Changes in snow mass contribute alone to more than 5 cm of amplitude, which represents more than 74% of the TWS variations north of 40°N. The change of the water mass (liquid and ice) represents over 34% of seasonal amplitude of total high latitude area-average TWS. The water reservoir from runoff (fast runoff and streams) contributes on average only 8 and 4% respectively, while the reservoirs related to floodplains and deep drainage are negligible (see Fig. 8). Since the TWS variations are mainly caused by snow mass changes and change of humidity of the soil, the impact of the inclusion of peatland processes has to be studied.

The contribution of each of these reservoirs is similar across the Ob, Lena and Yenisei basins. For these basins, the seasonal variations of TWS are higher than for the mean high latitude land surface with an amplitude of 13, 9.7 and 11 cm for the Ob, Lena and Yenisei basins, respectively. However, the simulated seasonal variations of TWS are strongly underestimated and represent only 65 %, 35 % and 43 % of the observed variations of TWS for the Ob, Lena and Yenisei basins, respectively.

Since snow mass changes represent three-quarters of the total change of northern latitudes TWS, the variations of modelled TWS is improved by the snow scheme in ORCHIDEE. The soil freezing process reduces the loss of water mass associated with snow and soil moisture during periods of frost. The soil freezing has a negligible impact on the other reservoirs, and is not responsible for the weak variations of the three routing reservoirs.

The impact of the inclusion of peatlands on the variation of TWS was studied at north of 40°N. We first study the impact of peatland on the variation of TWS, considering all grid cells where the fraction of peatland is non-zero (Fig. 8 b). We also study the change of TWS where the contribution of the SOIL reservoir is from the peatland column of soil only (Fig. 8 c).

Due to the small fraction of peatland at the northern latitude scale, the impact of the peatland scheme is negligible when we consider all the grid cells (Fig. 8 b) with a reduction in TWS changes of an amplitude of 0.10 cm due to the lower variation of soil humidity with the peatland scheme, where the inter-annual variations are low. The amplitude of TWS change of northern peatland grid cells only is higher than in all grid cells, reaching an increase in mass gain of 2 cm in 2009. The peatland scheme induces a reduction of 6% of the minimum of water loss. Moreover, the inter-annual variability of TWS in the regions of northern peatlands only is better represented than for the whole boreal regions.

The water budget of boreal peatlands was studied taking into account each of the aforementioned reservoirs with the exception of soil moisture where only the soil moisture of the peatlands is considered.

The water budget of boreal peatlands was studied taking into account the soil moisture on the column of peat only and each of the aforementioned reservoirs with the exception of the soil moisture of all column of soils. The corresponding TWS changes are shown in Fig. 8 c. The inclusion of the peatland scheme induces a reduction of 65 % of the annual TWS change of



these soils. The annual mean amplitude of the change of humidity of the peat soils only (Fig. 8 c) has been reduced by 1.26 cm (from 1.94 to 0.68 with the peatland scheme). The TWS change of peat soils has a mean annual amplitude of 6.77 cm against 8.17 cm when the scheme of peatland is not activated (STD). This leads to small inter-annual variations. The positive anomaly of the change of mass in 2002 amounts to only 3.8 cm with the simulation PEAT, against 5.4 cm with STD. This generates a

5 positive contribution of only 58 % for this year instead of 67 %. Conversely, the inclusion of the peatland scheme leads to a lower negative contribution of TWS from 58 % to 45 % in 2012, where the decrease in TWS reaches a minimum of -2.8 cm instead of -4.0 cm. This leads to a smaller seasonal variation where the amplitude is reduced on average by 1.0 cm.

In summary, the inclusion of the hydrology of peatlands leads to weaker water loss, which reduces the annual variation of the TWS of these soils, and to negligible inter-annual variability. Since these soils represent a small proportion of soils in northern

latitudes, these changes do not significantly affect the large-scale average TWS in northern latitudes.

### 3.4 Evaluation of modelled flooded peatland extents

The hydrology of peatlands in northern latitudes is difficult to assess on a large scale since the measurements are scarce. One

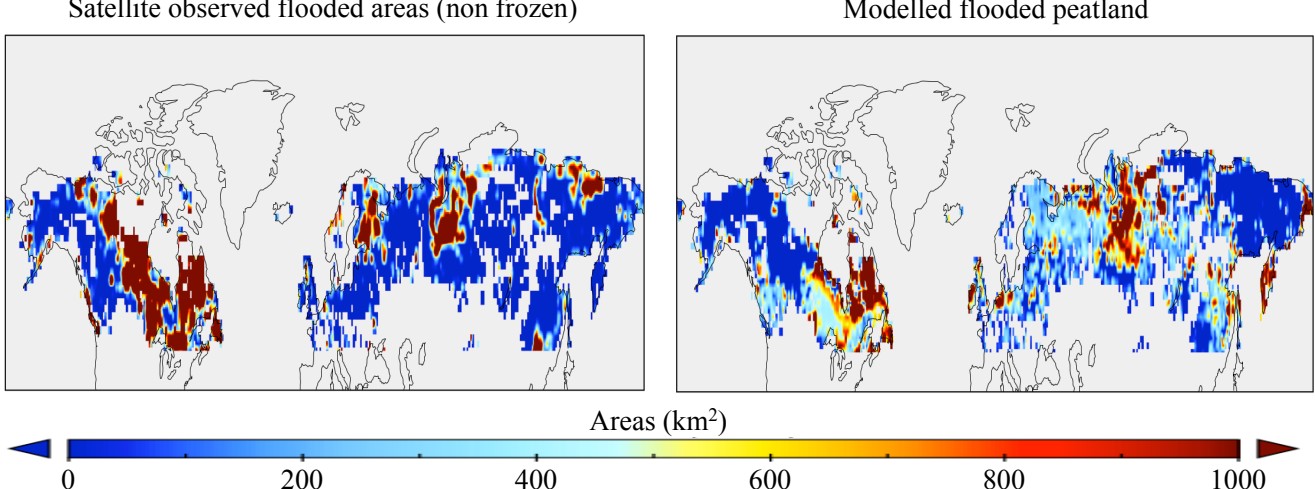

**Figure 9.** Extent of flooded areas of 0.5° grid cell in June monthly mean during the 1993-2006 period from satellite observations (left) and from simulated flooded peatlands (right).

option is to compare flooded peatlands with satellite observations of flooded areas by Prigent et al. (2007). These data come from a multi-satellite method for calculating flooded grid fractions at 0.25 °x 0.25 °monthly resolution of free-surface waters

for the period between 1993 and 2006, which have been interpolated on grid cells of $1° × 1°$ resolution. Here, we evaluate the location and the seasonality of flooded peatland using the CRUNCEP forcing of $1° × 1°$ resolution and with monthly outputs covering the same period 1993-2006. The average seasonality of the simulated and observed areas of flooded areas are compared with an average climate of the 1993-2006 period. Satellite observations can not capture frozen flooded areas. The comparison between observation and modelled flooded peatland was thus carried out by counting only the areas free of snow





and ice on the surface of the soil. Since satellite observations account for all the free water surfaces, these data are counted only when the observed flooded areas coexist with the grid points with a non-zero peatland fraction. Here, a peatland is considered flooded when the monthly average water table position is between the soil surface and 10 cm above the surface.

Fig. 9 shows the observed and simulated flooded surface areas during the month of June, when the fraction of frozen soil is
the lowest with a large extent of flooded area. The geographical distribution of flooded peatlands in the mean June is mainly located in eastern Canada and central Russia. Satellite observations of Prigent et al. (2007) also show a large area of free-surface waters in these areas as shown in Fig. 9. The spatial variability is well represented with the exception of western Canada and eastern Russia, where the model does not simulate flooded areas in June.

The flooded surfaces observed by satellite are present from April to August for Siberia and until September for Canada.
Concerning modelled flooded peatlands, the seasonality is lower, especially in the northeastern region of Canada where peat-lands are flooded throughout the year. Conversely, the extent of the modelled flooded peatlands is in sharp decline during the summer in western Siberia (not shown).

We compared the seasonality of flooded peatlands with satellite observation by considering only flooded peatlands when the soil is not frozen and in the absence of snow. Fig. 10 compares the observed seasonality among modelled peatland regions with
the modelled flooded peatlands in the absence of snow or freeze. For each grid point, it is assumed that the observed flooded fraction corresponding to the peatlands can not exceed the modelled fraction of peatlands. The total area is based on the size of the grid cell and the corresponding peatlands fraction obtained.

According to observations, the extent of flooded areas increases from April following the melting of snow. The increase in the modelled extent of flooded peatlands is less pronounced and occurs one month later. The simulated total extent of flooded
peatland reaches 0.55 Mkm$^2$ in June whereas the observations estimate this extent to 0.93 Mkm$^2$. The underestimation of flooded peatlands is probably due to the over-estimation of snow sublimation, known in the version of the model used and already mentioned above.

In observations, the area of flooded peatlands is maximum between May and August. Precipitation amounts are also signif-icant in summer in boreal flooded areas. As a result, the extent of flooded peatlands is well represented in July when summer
precipitation occurs.

Underestimation of flooded peatlands in spring occurs in Canadan and Siberian regions with high peatland fractions. In Siberia, the observed flooded areas are more concentrated in the center of the region, while the simulated flooded peatlands are more uniformly distributed over the entire region. For this study area, the modelled flooded peatlands are underestimated by 0.04 Mkm$^2$ in summer compared to observations, which means that only 66 % (Fig. 9) of the total area of this region is
represented by the model. In eastern Canada, the area of simulated flooded peatlands corresponds to only 88% of flooded areas observed in July-August. The model correctly simulates the flooded peatland areas near Hudson Bay and Quebec, but underestimates the flooded peatland areas to the south of Hudson Bay.

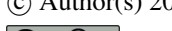



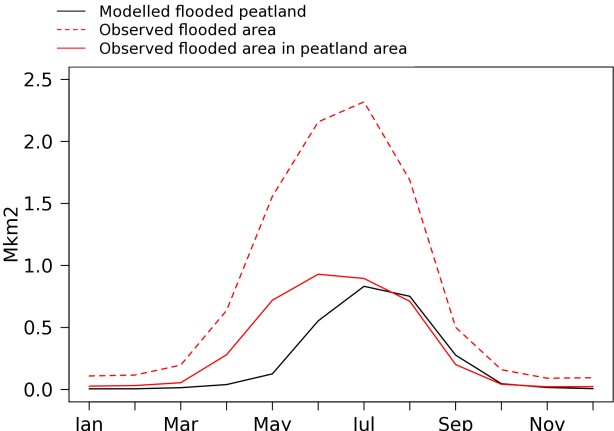

**Figure 10.** Mean seasonal cycle of flooded land areas of 0.5° grid cell for the 1993-2006 period from the modelled flooded peatland (black), observed flooded area above latitudes 45 °N (dashed red line) and from observed flooded area located in grid cells with non-zero fraction of peatlands (red line).

## 4 Discussion

This study has focused on northern peatlands since the majority is located at these latitudes where the increase of temperature is the most important. These peatlands are also often located in permafrost regions that are particularly sensitive to climate warming. We aimed at a better representation of hydrology of peatlands to account for their high carbon content in the soil and
to include them in global climate models. These developments can be used to estimate current and future associated methane emissions and to understand the sensitivity of hydrological variations to methane emissions from these northern peatlands.

The developments related to the vegetation present in peatlands remain however limited. In this study, we suppose a vegetation of peatlands close to the PFT grass already included in the model, with a lower root depth and a lower productivity. The thermal insulation of mosses which is particularly present in peatland is not taken into account.

The hydrological processes of the peatlands were mainly based on the significant water supply of peatlands from runoff with negligible drainage and stagnant water on the soil surface. The hydraulic properties of peat differ greatly from those of mineral soils. Peat soils have a high soil porosity with an important hydraulic conductivity near the surface, which decreases much faster than mineral soils (Letts et al., 2000).

Although these parameters have been measured for peat and parameterised in the model from the studies of Letts et al.
(2000); Dawson (2006); Beringer et al. (2001), showing high variations between sites and between sapric and fibric peat, we did not implement peat-specific hydraulics due to the configuration of the model which used the dominant soil texture per grid cell. This approximation has the effect to reduce the infiltration compared to real-world peat, and under-estimate the total water holding capacity of peat soil columns. One of the possible improvements is to define these properties according to the different soil columns, in order to improve the hydrological scheme of peatlands soils and other soils.





The water supply of peatlands that comes from the surface runoff of the other soils depends on their fraction within the grid cell. As a result, the water supply is higher the smaller the fraction of peatland within the grid cell. This phenomenon makes the peatland hydrology dependent on their fraction in the grid cell of the chosen resolution.

The site evaluation showed that the NEE is in agreement with observations and the internal hydrology of the model with a negligible deep drainage and a water stagnant above the surface allows to represent the water table profiles on peatland site when the local weather data are known. The differences in the water table profile between observations and simulations can come from the water supply of the runoff which is not known and not represented in the simulations on site. The model underestimates the WTD in winter for the Siikaneva and Fajemyr site, which can be explained by an overestimation of snow sublimation known in this version of the model.

The development of peatlands in the model does not distinguish the different types of peatlands, since this information is not known on a large scale. In this scheme of peatlands, we have chosen to feed the peat soils by surface runoff in addition to precipitation in order to obtain shallow WTD, which corresponds to the type of minerotrophic peatlands. This limitation enhances an overestimation of the WTD of ombrotrophic peatlands, which are not represented in this study. However, we located areas where, according to ORCHIDEE, peatlands could subsist as ombrotrophic bogs, in order to evaluate the sensitivity of these types of peatlands to precipitation. The results showed that the WTD of minerotrophic peatlands are more sensitive to precipitation than ombrotrophic peatlands. Areas where ombrotrophic bogs may exist on large scales (that is, under the applied large-scale meteorologicl forcing) are also less sensitive to precipitation than other regions because the weather conditions are sufficient to supply wetlands with water. These results were obtained with runs at the typical resolution of a climate model. At much smaller scales, variability of topography, meteorological forcing, soil parameters, etc., will of course enable ombrotrophic peatlands to subsist in areas where the large-scale conditions appear unfavourable.

The evaluation of the inclusion of peatlands in the model was carried out at different spatial scales. We studied the impact of this implementation which influences the routing of the surface runoff on the river discharge applied on the largest boreal watershed located in Siberia. We have found that hydrological processes of peatlands including the re-infiltration of surface runoff from other soils into peat soil induced a reduction of 20 % of the river discharge of the Ob basin. However, the reduction of the evaporation from these soils counterbalances this process. The modelled river flow remains under-estimated compared to the GRDC observations (Fekete et al., 1999), because the snow scheme used here underestimates the runoff in boreal regions. The new snow scheme in the more recent versions of the model makes it possible to improve the representation of the river flows of the boreal latitudes (Guimberteau et al., 2017).

The hydrological impact of this inclusion has also been studied on a large scale and at different time scales. We have shown that this inclusion has a negligible impact on terrestrial water mass variations, since the fraction of peatlands remains suffi-ciently small compared to the area of high latitudes. The variations of water mass linked to soil moisture of all soils contributes to 34% in average. This does not allow to influence the seasonal and inter-annual variations of the TWS in northern latitudes. However, the ORCHIDEE model in the standard version (STD) underestimates the inter-annual variability of TWS compared to the observations. In boreal regions, the inter-annual variability of TWS is best explained by the change of precipitation or



discharge (Landerer et al., 2010). A study with different meteorological forcing data could be made to better understand the bias due to the meteorological forcing used in this study.

The seasonal and inter-annual variations of water mass from the humidity of peat soils are reduced by 65% when the hydrological processes of peatlands are activated. Changes in snow mass contribute to an average of 74% of the total variation of TWS. These simulated variations are underestimated compared to the observations of the GRACE satellite (Tapley et al., 2004), which may be caused by the overestimation of the simulated snow sublimation. However, the modelled inter-annual variations of TWS in peatland grid cells are in accordance to those of TWS from observations. The inter-annual variations of TWS are better represented in peatland grid cells than for all grid cells of northern latitude above 40°N (Fig. 8).

At northern latitude scale, the modelled hydrology of peatland has been evaluated by comparing the area of modelled flooded peatland with the satellites observations from Prigent et al. (2007). This comparison showed that the areas where peatlands are flooded correspond to the flooded areas observed. Moreover, the extent of the observed flooded areas located in the grid points of the Yu et al. (2010) map and the simulated ones are equivalent in July for a simulated flooded extent of 0.83 Mkm$^2$ and observed from 0.89 Mkm$^2$. The snow scheme used in this version of the model enhances a negative bias during the spring where the modelled extent of flooded areas is underestimate from March to June. However, since this study was carried out by comparing modelled peatland from observations only in the grid cell where the fraction of peatland is greater than zero, this study is dependent on the peatland map used.

This study showed that the peatland scheme is consistent with observations both on site (Baldocchi et al. (2001); Schubert et al. (2010); Rinne et al. (2007); Lund et al. (2009)) and on a large scale (Prigent et al. (2007)). These developments have no strong impact on continental hydrology.

# 5   Conclusions

We have implemented peatlands in the high latitude version of the land surface model ORCHIDEE, in order to take into account their important role in the carbon and hydrological cycle. We have represented peatlands as a new PFT based on a global inventory peatland map from Yu et al. (2010). This study focused on the evaluation of the hydrology of northern peatlands which is crucial to represent the flux of carbon from these soils. The use of a fixed map makes it possible to follow the temporal evolution of the hydrological state of peatlands listed according to this map.

The hydrology of peatlands follows the internal hydrology of the model with a redistribution of the surface runoff from other soils, that is redirected into peat soils with a negligible deep drainage and a possible accumulation of water above the surface of these soils. These modifications are evaluated on site measurements that have shown the ability to represent the hydrological profile of peatlands.

We have shown that the reintroduction of the surface runoff of peatlands make these more vulnerable to changes in precipitation than peatlands fed only by precipitation. In this context, we considered that minerotrophic peatlands are more sensitive to precipitation than ombrotrophic peatlands. The location of ombrotrophic peatlands has a lower sensitivity to precipitation than minerotrophic peatlands areas.





The impact study of this implementation on the river flow and on the variation of terrestrial water storage showed that the incorporation of peatlands does not affect significantly the continental hydrology of the northern latitudes.

This study showed that the location and the seasonality of flooded peatlands are well represented, despite the low extent in early spring. The runoff of soils re-infiltrated into peat soils has resulted to a reduction of river flow of 20% continuously for the

growing season for the case of Ob basin. This reduction becomes negligible when the corrected evaporation flux is activated.

At the inter-annual scale, the variations of the modelled terrestrial water storage of northern latitudes is in accordance with the GRACE satellite observations. At these latitudes, the variations of mass of water from snow and soil moisture are the largest contributors, which represents respectively in average, from 2002 to 2014, 74% and 36% of the total variations of terrestrial water. The incorporation of peatlands induces a reduction, as well seasonally and inter-annually, of the variation of

soil moisture of peat soils. This reduction enhances a total reduction of 6% which can be neglected at this large-scale.

The new scheme represents peatland hydrology relatively well on a large scale, without disrupting the large-scale hydrology of the surface model and the hydrology of peatlands have small effects in the simulation of northern river discharge. This implementation will be further used to estimate the future evolution of the hydrology of peatlands and their associate methane emissions at the end of the century.

**Code availability**

The documentation and the code of the trunk version of ORCHIDEE are open source and can be found here: http://forge.ipsl.jussieu.fr/orchidee
The branch MICT-v1 (rev. 1255) of the code used in this paper with the development of peatland ORC-HL-PEAT (rev. 3058) can be found here: https://github.com/CLargeron/ORC-HL-PEAT/tree/MICT-PEATLAND. This path includes also the PFT map with peatlands which is needed to run the version ORC-HL-PEAT of the model. Readers who are interested to run this

model are encouraged to read the documentation of ORCHIDEE to understand how to use the model and to contact the corresponding author for further details on the peatland scheme.

*Acknowledgements.* This study has been supported by the PAGE21 project, funded by the European Commission FP7-ENV-2011 (grant agreement no. 282700) and the European Research Council Synergy grant ERC-2013-SyG-610028 IMBALANCE-P. Simulations with OR-CHIDEE were performed using computational facilities of the TGCC-Curie (CEA, France).



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
