# Peer review of "Implementing northern peatlands in a global land surface model: description and evaluation in the ORCHIDEE high latitude version model (ORC-HL-PEAT)"

_Geoscientific Model Development, 2017_

## Short Comment (SC1) · 18 Jul 2017

Dear Authors,

I'd like to thank you for making the code publicly available. However, Github branches are no permanent archives. Therefore the executive Editors of GMD highly recommend to make the exact code version, to which the paper referes to, available via an archiving system providing a DOI (e.g. Zenodo). In this way, the code version and the paper are perfectly linked to each other.

[Figure]

Best, Astrid Kerkweg

---

## Referee Comment (RC1) · Anonymous Referee #1 · 5 Feb 2018

**Implementing northern peatlands in global land surface mode: description and evaluation in the ORCHIDEE high latitude version model (ORC-HL-PEAT).**

This paper discusses the addition of peatlands into ORCHIDEE. There are two components for this – a vegetation component and a hydrology component. The vegetation component is mainly a modification of parameters. However the hydrology component involves the addition of some processes. These need to be described in more detail. I think more detail is required on the snow component which is frequently referred to in the results section but not discussed in the modelling section.

The vegetation is modified and there is a brief evaluation at the three FLUXNET sites. However it would be good to see the large scale effects of these changes. Also, since the snow seems to be a reason for many of the differences shown it would be good to maybe evaluate the snow component a little more.

It would be interesting to see a map of the observed peatlands and the different types.

I thin they authors should separate experimental design, observations used, model description and results more clearly. There are lots of slightly different experiments included, which is a bit confusing. There needs to be more consistency (particularly of terminology and experimental design) throughout the paper to make the story easier to understand.

Section 3.3.2 needs to be more focused on the key results given the question of how adding peatland affects the hydrology of northern high latitudes.

Minor comments

P2.L5 "The characteristics..." - sentence needs re writing.
P3. L14 What is the depth of the soil column and the approximate layer thicknesses?
P3. Section 2.2 – Paragraph 1 needs to go later. The first part of the section should discuss the site simulations, then the next part the large scale simulations. These should be more clearly separated. In addition and maybe more importantly, there is no experimental design for the large scale simulations despite several different experiments being included later.
P3. L1 There is no discussion of how the soil respiration is calcuated used for the NEE.
P6. L30 What hydraulic properties are used for peat soils?
P7. L3 – How much runoff from the other PFT soil columns?
P7. L7 How is the drainage blocked?
P7. Is there an equation or two that can be included to show how the hydrology was modified?
Fig 1. What is the depth of the soil?
P8. L1– Please rephrase this first sentence.
P8, L13 – any tests of the calibration of R for other sites? Maybe more details?
Fig 2. Looks a bit weird. In terms of the x-axis – there seems to be two days in a month. Results look very comparable though! The peatland pft has made a good change,
Fig 2. Do the observed NEE represent 100% peatland?
P9. L19. This says despite the missing processes we are getting the right answers? Surely this is unexpected?
P9. L22. Now it says Degero WTD is underestimated.....
Fig 3. Does Degero really have no winter precipitation? Maybe it is just too small to see? The axes could be adjusted?
Fig 3. I think these WTD results look pretty good for the summer. I think the winter differences need clear justification, see next comment.
Fig 3. Does the standard version of ORCHIDEE simulate a water table depth? If it does it would be

good to see how the improved representation of peatland affects this.

P10. L2. We have learned very little about snow up to now. Why cant the infiltration of snow occur?

Fig 4. How does this compare with observations?

Fig 5. How much does lack of snow infiltration affect things?

Section 3.2 This is a long section. Maybe I am missing the point but it seems to be telling us that minerotrophic peatlands are more sensitive to precipitation than ombrotrophic peatlands. This is expected? Is it the ability of the model to determine the difference between the two the novel result here? How much is this affected by the lack of representation of snow infiltration? It would be good to explicitly state the usefulness of these results for other applications.

P13. L10 This first paragraph is model description.

Fig 7. These subplots need a title. STD runoff is missing.

Fig 7. there is an interesting timing difference on the peaks between the observations and the model.

P14. L1 – Why is the STD version not the HL version aha it is (confusing!)?

P14. L3. This is the first time PEAT-LOWET has been introduced. It needs to go in the experimental design.

P14. L6 to P15. L5. This information can also go in the experimental design.

P15. L30 GRACE should be discussed earlier in a "materials and methods" section.

Fig 8 – What is HIGHLAT representative of – is this the same as STD? GRACE needs to be the same in each subplot. What do we learn from Fig 8a?

Fig 8. Do we learn anything from such a long time series or would a climatology for 1 year be better?

P16. L4 We have no "top panel" in fig 8.

Fig 9. I don't understand the units?

Serction 3.4 In section 3.2 there is quite a discussion of bog vs fen. Can this be referred back to in this section? How well does the model represent the flooded areas of bog vs fen? This would tie the relevant sections together.

Fig 10 – it seems the differences in timing in Figure 10 are caused by the snow. It might be good to show some snow results to demonstrate this?

---

## Referee Comment (RC2) · Anonymous Referee #2 · 1 Mar 2018

I believe the paper is worthy of publication once all the referee comments have been addressed to the satisfaction of the Editor.

Largeron et al. added a fixed gridcell fraction to represent northern peatlands to OR-CHIDEE. They then evaluated the impact of this new set of processes at site and regional scale.

In general, the arguments are quite convincing, but I feel they could be made much

more solid with the addition of some extra information (detailed below), and a little extra analysis. For I feel that this lack of detail in certain key places is the main drawback of the paper.

I have chosen to divide my comments into General and Specific categories in the following. The comments below refer, where possible, to specific pages and lines or ranges of lines.

I'm looking forward to reading a revised version.

General Comments and Questions
* * *
1. The problems with the treatment of snow sublimation, mentioned in many places throughout the text, is vital to be able to understand and trust the results shown here. Please provide more detail.

2. How are the peatland fractions of the cells parameterized? You mention that they have a texture, but is this the same as the remaining fraction of the cell, i.e. more representative of mineral soils? What is the porosity? What is the depth of the bucket? How is evaporation treated?

3. For the FLUXNET site evaluation: how was the WFDEI forcing combined with the climate at the site? What were the other components of the C flux, e.g. GPP, and do they compare to site observations?

4. Maps: Can you provide a map of your peatland fractions, described in Sec 2.3.1, or combine it with Fig 4? Can you highlight the Ob catchment on Fig. 4?

5. Productivity and decomposition: you reduce Vcmax based on Degerö observations, but how is the regional GPP affected by this change? Was the ORCHIDEE decomposition changed too? A lot of effort is put into a study of the effects on the regional water balance, but how do these peatland changes alter the regional C balance and fluxes?
[Figure]

How does the peatland vegetation behave, e.g. in terms of LAI, NPP, biomass etc.?

6. Little mention is made of the effect of the soil freezing and thawing processes. E.g. I suspect that the active layer depth is greater in your peatland cells than in reality, if the texture you use corresponds to mineral soils.

7. Section 3.2. Did you force the model with precip and NO runoff first in order to identify bogs and fens? After that, runoff transfer is switched off in cells with bogs? Please clarify.

8. Discussion. Can this approach be used to represent tropical wetlands? What is needed to do so? How about seasonally inundated wetlands? Will you add a CH4 submodel?

9. Language: Finally, please consider getting a native English speaker to go through your paper before it is published. The language is mostly fine, but there are quite a few places where the readability and grammar could be improved. I have identified some, but not all, below.

Specific Comments —————

Page 1: Line 3. The sentence "These are considered..." doesn't make sense. "These" are peatlands. Line 9. "at" or "for" instead of "according to"

P2: L3. Sentence "Moreover, ...." is too general, and anyway seems out of place here. L11. Same thing for "Peatlands..." L15. Remove "a large" L30. "peatland density"

P3: L4. "MICT" ??? L5. "latitude" L30. "as" to "to"

P4: L6. I suggest: "... with site measurements from the FLUXNET ..." L18. "shallow" vegetation? Do you mean low? L21. Västerbotten L28. remove final "of"

P5: L23: I suggest "in reasonable agreement with" instead L30-32: Mention that some models do have this stress, e.g. LPJ-WHyMe

P7: L4-7: I'm confused, was a simple bucket approach used, or did the inflitration etc follow STD ORCHIDEE? Eqns. 1 & 2. Do the Beta factors depend on vegetation type/PFT?

P8: - In general, I'd recommend you use "soil moisture" instead of "humidity" in the text. - R is described as a resitance and a reduction. - Eqn 3: so does this mean that when fpeat = 1, R = 1, so there is NO evaporation from the peatland? Is that justified? L10-12. show the improved latent heat flux!

P9: Fig.2 - is PFT grass simply the same simulation without the Vcmax reduction? L14. "snowfall and rainfall"

P10: L1. evidence for "may"? L1-2. unclear what's meant here. L3-5. But didn't you force Fayemyr with observed precipitation? If so, the WTP reduction on Feb/Mar should also be seen in the observations. Is the problem frozen soil? If so, where is the soil water coming from in reality? Is there justification for the sensitivity test described in Lines 9-10?

P11: Fig 5. Describe the zoomed section in caption.

P12: L1-2. Obvious? Sec. 3.3.1 - please identify the Ob basin in a Figure.

P13: Fig 7. There is no "center". Please identify with a, b and c, as in Fig 8. L8. "first show" - but both are in Fig 7!

P15: L10. "reduction" instead of "difference" L24. RE snow - more detail needed.

P16: L4. "top panel" - use (a) instead.

P17: L4. RE snow - more detail needed. L6 & L11. Fig 8a L14-17 - remove/move this paragraph? L22. Why north of 40N? I thought you ran the model for north of 45N. L18-29. This whole section is unclear.

P19: L1. ice on the observed or modelled soil? L12. Could also mention the discrepancy near Finland. L28-30. Not sure what this sentence means.

P20: L6. "methane emissions to hydrological variations", perhaps?

P21: L17. meteorological L31-32. Unclear "34% in average" of what?

P22: L8. Which Fig 8 - a, b or c? L14. underestimated

———————————————————————

---

## Author Comment (AC1) · 1 Apr 2018

**Response to Anonymous Referee #1:**

**Implementing northern peatlands in global land surface mode: description and evaluation in the ORCHIDEE high latitude version model (ORC-HL-PEAT).**

We thank the reviewer for his thoughtful comments. In the following, the reviewer's comments and suggestion are typeset normally, our replies to these comments and suggestions are written in **bold,** and the changes applied to the article are written in blue.

This paper discusses the addition of peatlands into ORCHIDEE. There are two components for this a vegetation component and a hydrology component. The vegetation component is mainly a modification of parameters. However the hydrology component involves the addition of some processes. These need to be described in more detail. I think more detail is required on the snow component which is frequently referred to in the results section but not discussed in the modelling section.
The vegetation is modified and there is a brief evaluation at the three FLUXNET sites. However it would be good to see the large scale effects of these changes. Also, since the snow seems to be a reason for many of the differences shown it would be good to maybe evaluate the snow component a little more.
It would be interesting to see a map of the observed peatlands and the different types.
I thin they authors should separate experimental design, observations used, model description and results more clearly. There are lots of slightly different experiments included, which is a bit confusing. There needs to be more consistency (particularly of terminology and experimental design) throughout the paper to make the story easier to understand.
Section 3.3.2 needs to be more focused on the key results given the question of how adding peatland affects the hydrology of northern high latitudes.

**To clarify the model experiments section, this has been split in 2 sections: Site simulations and large-scale simulations. The section 3.3.2 has been rewritten with fewer details and more clearly. We have added more details concerning snow component with explanations below:**

**In the version of ORCHIDEE-MICT used here, the snowmelt runoff bias due to snow simulation bias has been documented by Gouttevin et al (2012). Developments concerning a new representation of snow have been made with a multi-layer snow scheme (Wang_2013, Guimberteau_2017). However, this scheme could not be used when we started this study because it caused a non-conservation of water.**
**In accordance with the Reviewer 2, we have added more details concerning the explanation of the bias of the snow representation. The following sentences have been added:**
P3 L6-10 (in the modelling section): This scheme represents the changes of thermal and hydrological soil properties during periods of freezing and melting. This improves the latent heat exchange, water suction and the heat capacity depending on the ice content and the volumetric ice content (Gouttevin et al 2012}. The single-layer snow scheme in this model version supposes a constant snow density of 330 kg/m3 and is known to underestimate the snow cover depth as well as the snow water equivalent (Gouttevin et al 2012, Wang et al 2013}.
P17 L11: "The new multi-layer snow scheme from Wang et al (2013), Guimberteau et al (2017), not included in the model version used here, better represents snow depth and snow water equivalent (SWE), which were previously both underestimated in ORCHIDEE. This corrects the underestimation of snow melt runoff, and consequently improves the modelled river discharge in northern high latitudes (Guimberteau et al 2017)"
P17 L28: This trend is due to the same shift of the contribution of the modelled TWS, linked to the snowmelt that occurs about one month too early (Wang_2013).
P21 L6-8: The underestimate of flooded peatlands can be explained by an overestimate of snow sublimation as well as by an underestimate of the snow depth which leads to insufficient runoff from snow melt Gouttevin_2012b, Wang_2013).
P22 L30-32: "The model underestimates the WTD in winter for the Siikaneva and Fajemyr site. This can be explained by the overestimate of snow sublimation and the underestimate of the snow melt runoff known in this version of the model (Wang_2013)."
P22 L32-33: because the snow scheme used here underestimates the snow melt runoff in boreal regions (Wang_2013).
P23 L29-30: by the overestimation of the simulated snow sublimation and the underestimation of the

snow water equivalent in this version of the model (Wang_2013, Gouttevin_2012).

**Minor comments**

P2.L5 "The characteristics..." - sentence needs re writing.
Peatlands have specific properties concerning vegetation, hydrology as well as carbon.
P3. L14 What is the depth of the soil column and the approximate layer thicknesses?
**This information has been added L20 P3:**
The transport of water in the soil is described by the 11-layer scheme of  (DeRosnay et al 1999). The thickness of each layer increases geometrically with depth, from 1 mm in the top-soil to 1 m thickness at the standard 2 m total depth. Heat diffusion and moisture transport is calculated between each soil layer.

P3. Section 2.2 – Paragraph 1 needs to go later. The first part of the section should discuss the site simulations, then the next part the large scale simulations. These should be more clearly separated.
In addition and maybe more importantly, there is no experimental design for the large scale
simulations despite several different experiments being included later.
**The first part of the section explains the experimental design for large scale simulations. This has been put at the end of the section with the following introductory sentence:**
The impact of the inclusion of peatland in the model has been studied at large spatial scales, considering all northern peatlands above 45° North.
P3. L1 There is no discussion of how the soil respiration is calculated used for the NEE.
**It is unfortunately unclear to us what section this comment refers to.**

P6. L30 What hydraulic properties are used for peat soils?
**We have tested peatland-specific van Genuchten parameters as given in Table 1. We therefore write:**
P7 L31-32: We aimed at improving the representation the hydraulic properties of relevant large-scale peat soils by using appropriate Van Genuchten parameters of organic peat soils as described in Table 1.
**However, the hydraulic properties used in the model are unique for each grid cell, and the value used is based on the dominant mineral soil present in the cell. Therefore the hydraulic properties adapted for organic soils are not applied.**

P7. L3 – How much runoff from the other PFT soil columns?
**All the runoff is reinfiltrated into the peat soils. When this rate is too high, the runoff occurs in the next time step of the model with the column of soil of peatlands. Revised article:**
P8 L2: To represent these processes, we choose to infiltrate into the soil column of peatlands the entire runoff generated in the non-peatlands tiles of the same grid box at the same time step.

P7. L7 How is the drainage blocked?
**The flux of water at the bottom of the layer has been set to zero. This has been added to this sentence:**
P8 L7-8: To prevent water lost by the drainage, we choose to block the deep drainage at the deepest soil layer, with applying a zero-flux of the bottom drainage, because peatlands usually have no deep drainage.

P7. Is there an equation or two that can be included to show how the hydrology was modified?
**We have added two equations in the section 2.3.3:**

P8 L8: with applying a zero-flux of the bottom drainage ($q_N$= 0)

P8 L17-20: The water supply $W_{supply}$ of peat soils is summarised in the equation (Eq), $T_F$ is the throughfall, $S_{runoff}$ the runoff coming from non-peat soils and $R_{stagnant}$, the water from the reservoir.
$W_{supply} = T_F + S_{runoff} + R_{stagnant}$

Fig 1. What is the depth of the soil?
**The depth of the soil is 2 m as diagnosed in the standard configuration of ORCHIDEE. This information has been added in the caption of the Figure.**

P8. L1– Please rephrase this first sentence.

**This sentence has been modified:**
P9 L7: Peatland soils are flooded for part of the year, which leads to a vegetation saturated with water during this time.

P8, L13 – any tests of the calibration of R for other sites? Maybe more details?
**We have added in Fig 1b the turbulent latent heat flux for the Siikaneva site in accordance with the comments of the other reviewer.**

Fig 2. Looks a bit weird. In terms of the x-axis – there seems to be two days in a month. Results look very comparable though! The peatland pft has made a good change
**To improve the readibility, we have chosen to plot the diurnal cycle with a 10-day running mean filter. In fact, there is on average 3 composite diurnal cycles per month.**

Fig 2. Do the observed NEE represent 100% peatland?
**We consider peatland sites to be composed only by peatlands.**

P9. L19. This says despite the missing processes we are getting the right answers? Surely this is unexpected?
**Good point, the idea here is to show that it is almost sufficient to represent quite well the WTD. This can be due by a low water supply from runoff.**
**This has been changed as following:** Model results for the minerotrophic sites (Degero and Siikaneva) show that the water supply from precipitation only is almost enough to reproduce the observed water table position.

P9. L22. Now it says Degero WTD is underestimated.....
**This has been changed as following:**
Results from the Degero fens site slightly underestimate the WTD during the summer. This small bias...

Fig 3. Does Degero really have no winter precipitation? Maybe it is just too small to see? The axes could be adjusted? **Yes, the winter precipitation of Degero is not zero but very small. The axis have been chosen to be optimised for the 3 sites.**

Fig 3. I think these WTD results look pretty good for the summer. I think the winter differences need clear justification, see next comment.
Fig 3. Does the standard version of ORCHIDEE simulate a water table depth? If it does it would be good to see how the improved representation of peatland affects this.
**The standard version of ORCHIDEE does not simulate a water table depth. This water table depth can be used only for wetland areas since the standard depth of soil is 2 m. We have added the calculation of the WTD for peatlands. In the case of peat soils, the drainage flux at the bottom of the soil has been blocked since there is no deep drainage in peat.**
**See sentence P8 L7-8.**

P10. L2. We have learned very little about snow up to now. Why can't the infiltration of snow occur?
**Good point, this occurs when the soil is frozen. This sentence has been modified as follows**:
P11 L3: Since infiltration of snowfall is blocked when the soil is frozen, the water content is underestimated.

Fig 4. How does this compare with observations?
**Due to the lack of large-scale information on the distinction between peatland types, we have chosen to create a map in order to discern minerotrophic to ombrotrophic hydrological behaviour. The distinction between the type of peatlands could be made only with peatlands site.**
**We have added this information P11 L17-18:**
P11 L17-18: Since we cannot separate bogs and fens at the spatial scales relevant here, we consider in the model that all peatlands are fed by runoff. Due to the lack of large-scale information on the distinction between peatland types, we have chosen to create a map in order to discern the hydrological behaviour of the different type of peatlands. Due to the lack of large-scale information on the distinction between peatland types, we have chosen to create this map in order to discern the hydrological behaviour of the different types of peatlands.

Fig 5. How much does lack of snow infiltration affect things?
**To quantify the lack of snow infiltration, this study would need to be again evaluated with a version of the model that includes the new snow scheme. This is unfortunately beyond the scope of this study.**

Section 3.2 This is a long section. Maybe I am missing the point but it seems to be telling us that minerotrophic peatlands are more sensitive to precipitation than ombrotrophic peatlands. This is expected? Is it the ability of the model to determine the difference between the two the novel result here? How much is this affected by the lack of representation of snow infiltration? It would be good to explicitly state the usefulness of these results for other applications.
**Peatlands are very sensitive to precipitation. However, minerotrophic peatlands receive both direct precipitation and runoff, which indirectly depends on the precipitation of surrounding environments. This additional factor lets us suggest that minerotrophic fen could be more sensitive to precipitation. The lack of the representation of snow infiltration here could enhance an underestimation of peatlands considered as ombrotrophic bogs.**
**This has been added in the section:** Moreover, the total area of ombrotrophic bogs can be underestimated due to the lack of the snowmelt runoff in the version of the model (Wang et al 2013, Gouttevin 2012?).

P13. L10 This first paragraph is model description.
**First paragraph: The transport scheme of ORCHIDEE (Ngo-Duc et al., 2005) stores the water from the runoff and the drainage in 3 reservoirs with different residence times. Since the implementation of the peatlands in the model leads to the redirection of runoff from the other soil columns to the peat soils, one might expect some impact on the simulated river discharge. This impact is evaluated here for the Ob basin, which represents one of the largest boreal basins (above 45° N). This watershed is located in abundant peatland areas, particularly north of 60_ N. Although the average percentage of peatlands remains less than 10% per grid cell at 0.5° resolution, more than half of the grid cells have a non-zero fraction of peatlands. Above 60° N, peatlands are present on more than 96% of the grid points of the Ob basin.**
**The first sentence has been moved to model description of ORCHIDEE.**
**This paragraph has been modified and added to model experiments. Revised version:**
The impact of peatland on the river discharge has been evaluated for different catchments. Here, we present the results with the Ob basin, which represents one of the largest boreal basins above 45°° N. This watershed is located in abundant peatland areas, particularly north of 60N. Although the average percentage of peatlands remains less than 10% per grid cell at 0.5° resolution, more than half of the grid cells have a non-zero fraction of peatlands. Polewards of 60° N, peatlands are present on more than 96% of the grid points of the Ob basin.
**This sentence has been added to the second (becomes first) paragraph of the section 3.3.1:**
The implementation of the peatlands in the model leads to the redirection of runoff from the other soil columns to the peat soils. Here, we evaluate its impact on the simulated river discharge.

Fig 7. These subplots need a title. STD runoff is missing. **This change has been made in the Figure**

Fig 7. there is an interesting timing difference on the peaks between the observations and the model.
**It is unclear to us what this remark refers to. We see a maximum in June in both cases.**

P14. L1 – Why is the STD version not the HL version aha it is (confusing!)?
**To avoid confusion ORCHIDEE-HL has been added in the caption of Fig 7 and in following sentence:**
The standard simulation (STD) corresponds to the version of ORCHIDEE-HL which includes soil freezing \citep{Gouttevin_2012} and excludes the peatland scheme.

P14. L3. This is the first time PEAT-LOWET has been introduced. It needs to go in the experimental design.
**Thank you. This has been added in the hydrological processes section 2.3.3:**
In this study, we made two peatlands simulations: The first one, referred to as PEAT-LOWET in the following, includes the resistance to evaporation. In the second one, referred to as PEAT, no such resistance is applied.

P14. L6 to P15. L5. This information can also go in the experimental design.

**The first paragraph has been moved in the model experiments.**
**The GRDC observations is remind in the section as following:**
P16 L6: The GRDC (Fekete et al 1999) observed river discharge is shown as a dotted red line in Fig. 7a.

P15. L30 GRACE should be discussed earlier in a "materials and methods" section.
**This paragraph has been moved in the model experiments section.**

Fig 8 – What is HIGHLAT representative of – is this the same as STD? GRACE needs to be the same in each subplot. What do we learn from Fig 8a?
**The term HIGHLAT has been added in the caption:** variations of latitudes over 45° N (HIGHLAT).
**The importance of snow in TWS is shown in Fig 8a:** In the model, the accumulation of snow represents three-quarters of the total increase of TWS north of 45° N between Autumn and Spring.

Fig 8. Do we learn anything from such a long time series or would a climatology for 1 year be better?
**The long time series study was shown to avoid wrong conclusions due to an atypical year**.

P16. L4 We have no "top panel" in fig 8.
**This has been changed with adding a, b and c in the Figure**.

Fig 9. I don't understand the units?
**The units correspond to the peatland area (km2) per grid cell of 0.5° resolution. This has been added in the caption:** Extent of flooded areas (in km²) for each 0.5° grid cell…

Section 3.4 In section 3.2 there is quite a discussion of bog vs fen. Can this be referred back to in this section? How well does the model represent the flooded areas of bog vs fen? This would tie the relevant sections together.
**In the map, the flooded peatlands in summer are located in regions defined as bogs (because of the definition which say that WTD is <30 cm). This has been added P20 L15:**
The flooded peatlands areas coincide with the regions defined as ombrotrophic bogs (Fig. 4b). This can be explained by the definition of the bogs we used, that is, where their WTD does not exceed 30 cm depth for at least 4 consecutive months.

Fig 10 – it seems the differences in timing in Figure 10 are caused by the snow. It might be good to show some snow results to demonstrate this? **This is unfortunately not possible to demonstrate this. This study has started when the new snow scheme in ORCHIDEE was not implemented yet. A demonstration such this one requires to merge the peatlands scheme development in a version that includes the new snow scheme.**

---

## Author Comment (AC2) · 1 Apr 2018

**Response to Anonymous Referee #2**

We thank the reviewer for his thoughtful comments. In the following, the reviewer's comments and suggestion are typeset normally, our replies to these comments and suggestions are written in **bold,** and the changes applied to the article are written in blue.

Thank you for the opportunity to review the paper by Largeron et al.
I believe the paper is worthy of publication once all the referee comments have been addressed to the satisfaction of the Editor. Largeron et al. added a fixed grid cell fraction to represent northern peatlands to ORCHIDEE. They then evaluated the impact of this new set of processes at site and regional scale. In general, the arguments are quite convincing, but I feel they could be made much more solid with the addition of some extra information (detailed below), and a little extra analysis. For I feel that this lack of detail in certain key places is the main drawback of the paper. I have chosen to divide my comments into General and Specific categories in the following. The comments below refer, where possible, to specific pages and lines or ranges of lines. I'm looking forward to reading a revised version.

General Comments and Questions
————————————
1. The problems with the treatment of snow sublimation, mentioned in many places throughout the text, is vital to be able to understand and trust the results shown here. Please provide more detail.
**In the version of ORCHIDEE-MICT used here, the snowmelt runoff bias due to snow simulation bias has been documented by Gouttevin et al (2012). Developments concerning a new representation of snow have been made with a multi-layer snow scheme (Wang_2013, Guimberteau_2017). However, this scheme could not be used when we started this study because it caused a non-conservation of water.**
**These explanations have been added in P15 L24 and P17 L4 as specified in specific comments. Added also In:**
 P3 L6-10 (in the modelling section): This scheme represents the changes of thermal and hydrological soil properties during periods of freezing and melting. This improves the latent heat exchange, water suction and the heat capacity depending on the ice content and the volumetric ice content (Gouttevin et al 2012}. The single-layer snow scheme in this model version supposes a constant snow density of 330 kg/m3 and is known to underestimate the snow cover depth as well as the snow water equivalent  (Gouttevin et al 2012, Wang et al 2013}.
P17 L28: This trend is due to the same shift of the contribution of the modelled TWS, linked to the snowmelt that occurs about one month too early (Wang_2013).
P21 L6-8: The underestimate of flooded peatlands can be explained by an overestimate of snow sublimation as well as by an underestimate of the snow depth which leads to insufficient runoff from snow melt \citep{(Gouttevin_2012b, Wang_2013).
P22 L30-32: "The model underestimates the WTD in winter for the Siikaneva and Fajemyr site. This can be explained by the overestimate of snow sublimation and the underestimate of the snow melt runoff known in this version of the model (Wang_2013)."
P22 L32-33: because the snow scheme used here underestimates the snow melt runoff in boreal regions (Wang_2013).
P23 L29-30: by the overestimation of the simulated snow sublimation and the underestimation of the snow water equivalent in this version of the model (Wang_2013, Gouttevin_2012).

2. How are the peatland fractions of the cells parameterized? You mention that they have a texture, but is this the same as the remaining fraction of the cell, i.e. more representative of mineral soils? What is the porosity? What is the depth of the bucket? How is evaporation treated?
**The fractions of peatlands are inserted in the vegetation map of ORCHIDEE based on the peatland map of Yu et al. Concerning the properties of the soil, we have determined the properties of peatlands based on Letts et al 2000. However, the texture of organic soils that characterizes peat has not been taken into account since this version of ORCHIDEE used the dominant texture based on FAO. Therefore, peatland-specific textural parameters are applied only where peatlands are dominant. The depth of the soil is 2m and split in 11-layers (De Rosnay et al 1999).  The water balance is calculated for each type of soil where the evaporation of the peatlands soils is treated as a function of the soil moisture of the peatlands soils only. We then used a resistance factor R to**

reduce evaporation for peatlands soils to counterbalance the fact that the (assumed) water holding capacity and infiltration of mineral soils used in the model for peat is lower than the one of actual peat soils and that the ability of mosses to retain water is not represented.

**This explanation is described in the model description in section 2.1 as following (the sentence added is written in bold):**

The transport of water in the soil is described by the 11-layers scheme of (DeRosnay et al 1999), **where the layer thickness of each layer is increasing with depth, from 1 mm to 1 m depth with the standard 2 m total depth scheme.** The heat and moisture transport is calculated between each soil layer. The water balance of the soil is defined separately as a function of class of vegetation and clustered for bare soil, trees and grasses. Each of these three soil types has a separate water balance. The fraction of the area of each soil type is calculated as a function of the fraction of the area of the corresponding PFT. However, the soil porosity is defined only as a function of the dominant soil texture in the grid cell, based on textural classification data from the global Food and Agriculture Organization map (FAO, 1978). Only one soil parameter is defined per grid cell, which describes hydraulic conductivity, residual and saturated water content as well as the Van Genuchten parameters, which describe the hydrological properties of the soil. »

**In order to clarify this in the paper we have added the following sentence to remind this in the hydrological processes section of peatlands:**

P7 L3-6: "This leads to a separation of the water balance of peatlands soils that is crucial to represent the water content of these soils. The calculation of evaporation is therefore separated for these soils, where an adjustment can subsequently counterbalance the non-representation of the mosses often present in these environments, which have a significant capacity to retain water."

3. For the FLUXNET site evaluation: how was the WFDEI forcing combined with the climate at the site? What were the other components of the C flux, e.g. GPP, and do they compare to site observations?

**For the site evaluation, flux tower meteorology is used to prescribe precipitation to the model. However other meteorological forcing (wind, air humidity, pressure...) comes from WFDEI. The observed eddy covariane NEE flux is then compared to modelled NEE at each sites.**

**This sentence has been modified as following:** (P4 L27-29)

The site evaluation was performed using WFDEI meteorological forcing at the 0.5° grid cell containing each site, excepted for precipitation that was prescribed from flux tower observations (FLUXNET data).

4. Maps: Can you provide a map of your peatland fractions, described in Sec 2.3.1, or combine it with Fig 4? Can you highlight the Ob catchment on Fig. 4?

**In the Fig 4 are represented all the grid cells where the fraction of peatland is greater than zero.**

**A map of the fraction of peatland is added in Figure 4 with a highlight over the Ob catchment with the following title:** (a) map of peatlands fraction inserted in the model vegetation map at 0.5° resolution. The red domain highlights the Ob catchment.

**Added in the text:** P5 L21 peatland map... in order to obtain a fraction of peatland fpeat_Yu for each grid cell as shown in Fig. 4(a). P14 L7: This impact is evaluated here for the Ob basin (represented in red in Fig 4a)

5. Productivity and decomposition: you reduce Vcmax based on Degerö observations, but how is the regional GPP affected by this change? Was the ORCHIDEE decomposition changed too? A lot of effort is put into a study of the effects on the regional water balance, but how do these peatland changes alter the regional C balance and fluxes? How does the peatland vegetation behave, e.g. in terms of LAI, NPP, biomass etc.?

**Good point. Yes, we have applied a lower decomposition of active carbon for peatlands soils based on the soil moisture of the peatlands soils.**

**This paper has mainly focused on the hydrological effects of peatland and the changes made to carbon were not described in this paper. Another study has further improved the carbon budget of peat based with additional adaption of the peat hydrology described in our study (Qiu et al 2018).**

6. Little mention is made of the effect of the soil freezing and thawing processes. E.g. I suspect that the active layer depth is greater in your peatland cells than in reality, if the texture you use corresponds to mineral soils.

**The active layer thickness is reduced with organic soils compared to mineral soils. Unfortunately, this mechanism is not accounted for in our model.**

7. Section 3.2. Did you force the model with precip and NO runoff first in order to identify bogs and fens?

After that, runoff transfer is switched off in cells with bogs? Please clarify.

**Yes, that is right.**

**To model bogs we made a simulation switching OFF runoff transfer from other PFTs and defined peat in a grid cell as bog if the water table in this grid cell was not deeper than 30 cm for at least 4 consecutive months. After that simulation, we switched OFF runoff transfer for all bogs and other peat grid cells are defined as fen.**

**This has been added in the corresponding paragraph as presented below:**

"We modelled ombrotrophic bogs, i.e. peatland fed only by rainfall that do not receive input from other soil columns, in two steps. First, we made a simulation switching off runoff transfer from other PFTs and defined peat in a grid cell as ombrotrophic bog if the water table in this grid cell was not deeper than 30 cm (in accordance with observations \citep{Booth_2005}) for at least 4 consecutive months in the mean year, for the 1990-2010 period. After that simulation, we continued to switch off runoff transfer for all ombrotrophic bogs, while the water balance of all other peat grid cells defined as minerotrophic fen is simulated with runoff transfer being switched on. Usually the ombrotrophic bog condition tends to be fulfilled between January and April."

8. Discussion. Can this approach be used to represent tropical wetlands? What is needed to do so? How about seasonally inundated wetlands? Will you add a CH4 submodel?

**This approach may possibly be used for tropical peat but its limitation is that runoff is only received from PFT in the same grid. If a tropical peat system is connected to large-scale hydrological network with water routing connecting grid-cells, then our approach cannot be used. However, the validation and the evaluation have been done only at high latitudes and would need to be studied for tropical wetlands. This model might be better for tropical wetlands since there are no issues related to snow. However, the reduction of the evaporation might be not adapted for tropical conditions. We have adapted the model of methane emission from Walter et al 2001, which is very sensitive to the variation of WTD. The results concerning methane emission are not presented in this study.**

**This paragraph has been added at the end of the discussion:**

This approach may possibly be used for tropical peatlands but its limitation is that runoff is only received from PFT in the same grid. If a tropical peat system is connected to large-scale hydrological network with water routing connecting grid-cells, then our approach cannot be used. Moreover, this would need to be evaluated for tropical peatlands.

**We have added a CH4 sub-model adapted for peatlands, which takes into account the variation of the WTD. This has been made from the flux density model of Walter et al 2001 and the reintroduction of this model with simplification from Ringeval et al 2010.**

**To explain this we have added the following sentence:**

P21 L7-10: For this, an adaptation of the CH4 sub-model for flooded wetlands by (Ringeval et al 2010) has been implemented which takes into account the variations of WTD. It is based on the methane flux density model of Walter et al 2001 and used to evaluate the methane emission from peatlands as described in this study (Largeron, 2016). However, the methods and results of the methane emission are not shown in this paper.

9. Language: Finally, please consider getting a native English speaker to go through your paper before it is published. The language is mostly fine, but there are quite a few places where the readability and grammar could be improved. I have identified some, but not all, below.

Specific Comments —————–

Page 1: Line 3. The sentence "These are considered..." doesn't make sense. "These" are peatlands. These peatlands are represented as a new Plant Functional Types (PFT)

Line 9. "at" or "for" instead of "according to" at different spatial and temporal scales

P2: L3. Sentence "Moreover, ...." is too general, and anyway seems out of place here.

**This sentence has been deleted**

L11. Same thing for "Peatlands..." **This has been deleted**

L15. Remove "a large" : **This has been deleted in the text.**

L30. "peatland density" **This has been changed in the text.**

P3: L4. "MICT" ??? **ORCHIDEE-MICT used here is a preliminary version of Guimberteau et al 2017}**
**This has been clarified in the text**.

L5. "latitude" L30. "as" to "to" **changed in the text.**

P4: L6. I suggest: "... with site measurements from the FLUXNET ..." **This has been changed in the text.**

L18. "shallow" vegetation? Do you mean low? **Yes, low vegetation. This has been changed in the text.**

L21. Västerbotten **changed in the text**

L28. remove final "of" **deleted in the text**

P5: L23: I suggest "in reasonable agreement with" instead **This has been changed instead of in adequacy**

L30-32: Mention that some models do have this stress, e.g. LPJ-WHyMe

**In peatlands, the vegetation can survive in saturated areas.** The representation of inundation stress is taking into account in some models such as LPJ-WHy \citep{Wania_2009b}.

P7: L4-7: I'm confused, was a simple bucket approach used, or did the infiltration etc follow STD ORCHIDEE? Eqns. 1 & 2. Do the Beta factors depend on vegetation type/PFT?

**In this study, we used the standard 11 layers described by De Rosnay 1999. However, the WTD is diagnosed by taking into account all the water contained in each layer previously diagnosed, as if it were transferred to a bucket.**

**The parameter $B_{inter}$ and $B_{transpir}$ is calculated for each PFT. However, $B_{evap}$ is calculated only per grid cell. This explains why we applied the reduction factor R only in function of the fraction fpeat. To avoid confusion, the following sentences has been added** P7 L14: "In this study, we use the standard 11-layers scheme of ORCHIDEE as described by \citet{DeRosnay_1999} to represent the hydrology of peatlands."

P7 L35 P8 L1: The parameters $B_{inter}$ and $B_{transpir}$ are calculated for each PFT. However, the evaporation capacity $B_{evap}$ is calculated only in function of the grid cell

P8: - In general, I'd recommend you use "soil moisture" instead of "humidity" in the text. - R is described as a resistance and a reduction. - Eqn 3: so does this mean that when fpeat = 1, R = 1, so there is NO evaporation from the peatland? Is that justified?

**We choose a resistance R in accordance with the soil moisture. I changed reduction into resistance and soil moisture instead of humidity. When fpeat=1 (almost never the case) the evaporation is not reduced. This sentence has been added in the text:**

"The reduction of evaporation does not occurs when $f_{peat}=1$. "

L10-12. show the improved latent heat flux! **The figure has been added with the Figure 1. Caption**: (b) Observed (black) and modelled (green) turbulent latent heat flux (LE) in W/m2 of the peatland site of Siikaneva

P9: Fig.2 - is PFT grass simply the same simulation without the Vcmax reduction? **PFT grass doesn't have the Vcmax reduction and doesn't have the hydrological processes applied to the PFT peatland.**

L14. "snowfall and rainfall" **changed in the text**

P10: L1. evidence for "may"? **I have changed the following sentence:** This bias can be explained by the amount of water from groundwater, which is not represented in the model. The opposite is observed at the Siikaneva site and could come from an outgoing flow as a small drainage rate.

L1-2. unclear what's meant here. **This has been modified in the text as following:** "The opposite is observed at the Siikaneva site where the WTD is overestimated. This could come from an outgoing flow as a small drainage rate."

L3-5. But didn't you force Fajemyr with observed precipitation? If so, the WTP reduction on Feb/Mar should also be seen in the observations. Is the problem frozen soil? If so, where is the soil water coming from in reality? Is there justification for the sensitivity test described in Lines 9-10? **The modelled WTD of peatlands are diagnosed from the FLUXNET precipitation of this site. 2005 is a dry years and could explain why the modelled WTD is deeper than observations. In reality, there might be different properties of the soil as well as presence of mosses that can hold water. This has been added in the text:**

"The specific vegetation at the Fajemyr site such as mosses helps to hold the water which can then be infiltrated into the soil. The underestimation of the modelled WTD can be explained in part by the absence of the representation of mosses in the model."

P11: Fig 5. Describe the zoomed section in caption. **Sentence added to the caption:**

The region of western Siberia (zoomed area) has been used for a sensitivity study of peatland to precipitation.

P12: L1-2. Obvious? Sec. 3.3.1 - please identify the Ob basin in a Figure. **This has been added in Fig 4a**

P13: Fig 7. There is no "center". Please identify with a, b and c, as in Fig 8. **This has been changed**

L8. "first show" - but both are in Fig 7! **This sentence has been deleted**

P15: L10. "reduction" instead of "difference" **changed in the text**

L24. RE snow - more detail needed. **This has been added in the text: (now P17-L11)** "The new multi-layer snow scheme, not included in the model version used here, better represents snow depth and snow water equivalent (SWE), which were previously both underestimated in ORCHIDEE. This corrects the

underestimation of snow melt runoff, and consequently improves the modelled river discharge in northern high latitudes (Guimberteau et al 2017)"

P16: L4. "top panel" - use (a) instead. **This has been changed**

P17: L4. RE snow - more detail needed. **This has been added in the text**: (now L28-29 P17)

"This trend is due to the same shift of the contribution of the modelled TWS, linked to the snow melt that occurs about one month too early \citep{Wang_2013}."

L6 & L11. Fig 8a **This has been added.**

L14-17 - remove/move this paragraph? **This has been removed.**

L22. Why north of 40N? I thought you ran the model for north of 45N. **For this study, each reservoir has to be calculated explicitly. The routing scheme in the model is used here. However, the routing scheme cannot run at 45°N because the scheme need information from the whole river basin information.**

L18-29. This whole section is unclear. **This whole section has been modified as described below:**

Modelled TWS in Fig 8a has the same phase than in the GRACE observation, but the annual amplitude is underestimated by 27% in the model. The simulation of TWS is improved by the soil freezing and snow parameterization introduced by Gouttevin et al (2012). In the model, the accumulation of snow represents three-quarters of the total increase of TWS north of 45°N between Autumn and Spring (blue curve in Fig 8a). In addition, the soil freezing parameterization keeps a mass of water stored as ice in the soil pores (green curve in Fig 8a) instead whereas in absence of freezing, liquid water losses from runoff and drainage would decrease TWS. By contrast, the three free water reservoirs corresponding to water being routed to the ocean (red, orange, pink curves in Fig 8a) have a small seasonal variation and thus do not contribute to the amplitude of TWS.

In our model, peatlands store a fraction of runoff water that is not transported to the ocean.

To evaluate the impact of peatlands on the variation of TWS, we selected only the grid-cells containing some peat (non-zero peat fraction) and performed two simulations. We have evaluated the contribution of the SOIL reservoir both for all column of soil (Fig. 8 b) and for the column of soil of peatland only (Fig. 8 c). The results shown in Fig. 8b indicate that the additional storage of water contributed by the peat fraction in the selected grid cells is negligible (the curves with and without peatlands are merely distinguishable).

In this case, the TWS changes are reduced by 0.10 cm due to the lower variation of soil humidity with the peatland scheme, where the inter-annual variations are low. Moreover, the inter-annual variability of TWS in the regions of northern peatlands only is better represented than for the whole boreal regions.

However, the TWS change in northern peatlands is higher when only the soil column of peatland is considered (Fig. 8c) and reaches an increase in mass gain up to 2 cm in 2009. However, more water storage only occurs in the column of peat soil and not in the other column of soil of a grid cell. When the column of peat soil only is considered (Fig. 8c), the TWS change is higher and reaches an increase in mass gain up to 2 cm in 2009. This is the reason why we show in Fig 8c simulated TWS averaged over the area covered by peat only (column of peat soil) compared to a simulation where the same area is covered by grass. Local TWS remains lower in summer with peat than with grass, with a mean difference of 1.30 cm.

P19: L1. ice on the observed or modelled soil? **Modelled soil: Since satellite observations do not retrieve flooded areas being frozen we consider flooded area during the thaw season.**

Revised text: P20 L5: "The comparison with satellite flooded area was restricted to the thawing season, because of no satellite retrieval of flooded area being frozen »

L12. Could also mention the discrepancy near Finland. **This has been added in the text.**

L28-30. Not sure what this sentence means. **This sentence has been modified:** « In western Siberia, the model underestimated the flooded peatlands of 0.04 Mkm2 in summer compared to observations. This means that the model represents only 66% of the observed extent of flooded peatland in this region »

P20: L6. "methane emissions to hydrological variations", perhaps? **Yes, that's right. This has been changed in the text.**

P21: L17. meteorological **This has been corrected in the text**

L31-32. Unclear "34% in average" of what? **This sentence has been modified:**

"The variations of water mass linked to soil moisture of all soils contribute to 34\%\ of the annual mean TWS."

P22: L8. Which Fig 8 - a, b or c? **Comparison between Fig 8 a and b:** (Fig. 8 (a) and (b)).

L14. Underestimated. **This has been corrected in the text**

---

## Author Response (AR3)

**Response to Anonymous Referee #3 (June 2018):**

**Implementing northern peatlands in global land surface mode: description and evaluation in the ORCHIDEE high latitude version model (ORC-HL-PEAT).**

*We thank the reviewer for his thoughtful comments. In the following, the reviewer's comments and suggestion are typeset normally and our replies to these comments and changes applied to the article are written in* blue.

*General comments*
*This paper is some what improved however I feel that there are still some outstanding issues to be addressed and with some careful thought on slight restructuring and simplification it can still be significantly improved.*
*I feel the structure needs to be modified. In particular the modelling improvements (2.3.2 and 2.3.3) need to come directly after the model description so the reader hasn't forgotten the structure of the original model.*
*The modelling improvement section has been moved before the model experiments section. The sentences below have been added in this section:*
*P4 L1-3: This section describes the developments and methods used to incorporate peatlands into the model.*
*P5 L18: see site description section 2.3.1*
*L23: (see section 2.3.1 for site description).*
*Section 2.3.2. could be made clearer on exactly what has been modified for the peat pft – just two things? To be clearer, we have added the following sentence at the end of the section:*
*The new PFT peatland corresponds to a flood tolerant C3 grass with the properties of the PFT C3 grass, where we applied a lower productivity due to the lack of nutrient and with a reduction of the rooting depth.*
*The stated aim of the paper is to look at the hydrology which seems to have compensating errors so on the large scale there is a small net effect. However Figure 2 is definitely not hydrology and actually the pft changes are good. It is actually a really good result, so should probably stay in but then that still begs the question of what about the large scale nee/gpp? We have reduced the carboxylation rate in order to better fit with observations. The comparison of modelled and observed NEE at large-scale remains however difficult to evaluate. Therefore, we have chosen to compare the peatland NEE with other PFT.*

*Also the bit about CH4 in the first paragraph of the discussion makes me want to ask more about it. I am not sure that the second part of the paragraph is that relevant here.*
*This paragraph has been added due to the request of a reviewer. We have reduced the length of this paragraph and this has been moved at the end of the discussion.*

*The site descriptions can be put in a table to simplify things. We have added a table to summarise the site descriptions: Table 2: "Descriptions of peatlands sites used for site evaluation. The "Years" column corresponds to the available years of FLUXNET meteorological data.*

*I presume the soils are not compressible? What impact might including those have on the results?*
*No, the soils are not compressible on seasonal time scales. The representation of the compressibility of soils must be carried out in the case of a study over long time scales. The time scale used in this study is not really long enough to warrant that this process be represented explicitly.*

*Do you have any suggestions as to how to deal with the issue of resolution dependence? How might this impact the TWS results? The PFT map is re-interpolated into the resolution of the model and allows a different percentage of each PFT as a function of the size of the grid cell. Concerning*

*peatlands, the water supply is also dependent of the resolution. On the scales of interest here, peatlands mostly represent a small fraction of the grid, and moderate resolution increases will not necessarily lead to very strong variations of the peatland fractions at the grid scale.*
*The corresponding paragraph in the results section has been modified as follows:*
*"The water supply of peatlands that comes from the surface runoff of the other soils depends on their fraction within the grid cell. As a result, all other things being equal, the water supply increases as the fraction of peatland within a grid cell decreases. In an extreme case, in which peatlands would be spatially concentrated in a small area within a larger region (because of topographic constraints, for example), this phenomenon would make the peatland hydrology resolution-dependent; however, the large-scale hydrology would certainly also be resolution-dependant in such a case. »*

*Can you cite some literature to suggest that we expect peat soils to have reduced variability of water storage?* *The reduction seen in the model is negligible:*
*P25 L31: This reduction enhances a total reduction of 6 % of the total variations of terrestrial water, which can be neglected at this large scale.*
*In the literature, this trend is uncertain and depends of the peatlands sites (Bullock and Acreman, HESS, 2003, doi: 10.5194/hess-7-358-2003).*

*Also is there some evidence in the literature that minerotrophic peatlands might be more sensitive to precipitation than ombrotrophic peatlands in the real world?* *Minerotrophic peatlands receive both direct precipitation and runoff, which indirectly depends on the precipitation of surrounding environments. This additional factor lets us suggest that minerotrophic fen could be more sensitive to precipitation. Unfortunately, we didn't find other studies on this topic.*

*How different is the precipitation forcing between WFDEI and CRU-NCEP?*
*In this study, we have compared the river flow of the Ob basin with using two meteorological forcing in order to know the sensitivity to the meteorological input. We have used WFDEI and CRUNCEP forcing, where the precipitation in the CRUNCEP dataset is slightly higher in average along the Ob basin than WFDEI. In average, the precipitation with WFDEI forcing enhances a peak of runoff one month earlier than with CRUNCEP forcing. The resulting modelled river flow is not really sensitive to the meteorological forcing such as seen in Wu et al (2018) and we have chosen to illustrates the difference of mean runoff instead of the mean precipitation over the Ob basin.*

*Please can you get a native english speaker to read the final version through – there are some awkward phrases particularly in the introduction, but also throughout the text.*
*Yes. The changes were made in this paper. We hope that the revised version will be satisfying.*

*Minor comments*
*Page 3, line 25 – one soil parameter is misleading.* *This has been changed into: one soil texture*
*Page 6, first paragraph not relevant. Also I am not quite sure of the meaning of the last sentence in section 2.3.1. Not all of the peatlands observed have been identified?* *The peatlands have been added at the expense of the fraction of PFT grasses only.*
*This sentence is confusing and has been removed.*
*Section 2.3.2 line 36 is repeated information.* *I don't see any line 36 in section 2.3.2*
*Page 8 line 1 – lateral water flow comes from runoff and indirectly from precipitation. I am a bit confused about whether there is any difference in treatment between surface and subsurface runoff.*
*This has been changed: Peatland water inflow comes from precipitation, surface runoff and from nearby soils. In the case of peatlands, we have blocked the sub-surface runoff. The water inflow*

*corresponds to the direct precipitation and from the surface runoff to other soils which is redirected into peat soils.*

Page 10 line 31 – what about surface runoff? Why cant we add simulated subsurface runoff?
*Since we cannot measure the observed sub-surface runoff flows, we couldn't consider these flows in the model. The surface runoff represents the precipitation and the redirected runoff from other soils.*

Page 11 line 4 – these three sites? Just talking about two sites in the previous sentence.
*We have changed these three sites into "For the 3 sites of this study"*

Why are the differences in the Siikaneva not snow related?
*The difference in winter is related to a poor representation of the snow scheme, where the percolation of water is not taken into account. In this study, the single-layer snow scheme is used. When the soil is frozen, the modelled water flow towards the soil is blocked. For the Siikaneva site, snow cover starts in early December until April. This coincides with the period when the modelled WTD is underestimated. We changed the explanation in the text as follows:*
 *"The opposite is observed at the Siikaneva site, where the WTD is overestimated during the summer, which could come from an outgoing flow  such as a low drainage rate.*
*In winter, the modelled WTD is underestimated when the soil is frozen.  The simple-layer snow scheme used in this study does not represent percolation of water. When the soil is frozen, the infiltration of water is blocked, which leads to underestimate the water content in the soil leading to an underestimation of the WTD."*

*Figure 1b is separate from Fig 1a so should be separated. Also figure 4a can be separated and made figure 1.  These Figures have been separated.*
I still think figure 2 looks a bit odd. Aso toy want the new model results to stand out not the old ones.  *The Fig. 2 (now Fig. 4) shows the impact of the reduction of the productivity applied for the PFT peat compared to the original PFT grass and this match well with the observed NEE of 3 peatlands sites. This type of figure using a 10 day running mean is frequently used to better describes the mean daily profile of the NEE free of the daily perturbations (such as Krinner et al 2005,  Suyker et al 2003, Ukkola et al 2017).*

*Figure 3. change axes for precipitation.  For the site of Degero, the observed precipitation is unknown in winter. We have chosen this axe for a question of visibility and a comparison for the 3 sites of peatlands.*

*Figure 7 – the differences between the dashed and solid line need to be more clearly defined in the caption. The caption has been changed as following: "Mean annual cycle of river discharge, (b) runoff of peat soils and (c) evapotranspiration of peatlands, for the Ob river basin. The observed river discharge is represented in red. The simulation are given with the standard version of ORCHIDEE-HL (STD) (black), with peatlands scheme without (PEAT) (blue) and with the reduced (PEAT-LOWET) (green) evaporation, using the meteorological forcing CRUNCEP (full line) and WFDEI (dashed line).*

*Figure 8 still ahs too many lines and too many seasonal cycles on it to see it clearly. There is an interesting time offset in the TWS between the model and observations which will become clearer when the x-axis is expanded. Move (b) in the figure caption to the beginning of the sentence.  This has been changed as following: (b) including the water reservoir variations of total soil moisture of the soil and (c) the variations of total soil moisture of the peat soil column only.*
Please change humidity to soil moisture. *This has been changed.*
STD is used twice for two different simulations.  *The term STD corresponds to the simulation where the properties of the PFT peat are not taken into account (i.e. corresponding to PFT grass properties with standard hydrology).*
Page 20 line 27 – how did you define the presence or absence of frozen soils and snow – this is going to be different depending on the model or observational data set, so it is hard to make the two

*comparable.* *Yes of course. In this study, we applied the comparison when the amount of the modelled snow is set to zero and when the surface temperature of the soil is above 0°C.*

*Page 22, line 15 – this sentence or two can probably be removed.* *This has been added following the request of a reviewer. This paragraph has been reduced as following:*
*For this, an adaptation of the CH4 sub-model for flooded wetlands by Ringeval et al (2010) and based on the methane flux density model of Walter et al (2001) has been implemented taking into account WTD variations of peatland soils (not shown in this paper) (Largeron et al 2016).*